# MLL-fusion-driven leukemia requires SETD2 to safeguard genomic integrity

Anna Skucha[1,2], Jessica Ebner[2], Johannes Schmöllerl[2], Mareike Roth[3], Thomas Eder[2], Adrián César-Razquin[1], Alexey Stukalov[1], Sarah Vittori[1], Matthias Muhar[3], Bin Lu[4], Martin Aichinger[3], Julian Jude [3], André C. Müller[1], Balázs Győrffy[5], Christopher R. Vakoc[4], Peter Valent[6], Keiryn L. Bennett [1], Johannes Zuber [3], Giulio Superti-Furga [1,7] & Florian Grebien [2,8]

MLL-fusions represent a large group of leukemia drivers, whose diversity originates from the vast molecular heterogeneity of C-terminal fusion partners of MLL. While studies of selected MLL-fusions have revealed critical molecular pathways, unifying mechanisms across all MLL-fusions remain poorly understood. We present the first comprehensive survey of protein–protein interactions of seven distantly related MLL-fusion proteins. Functional investigation of 128 conserved MLL-fusion-interactors identifies a specific role for the lysine methyltransferase SETD2 in MLL-leukemia. SETD2 loss causes growth arrest and differentiation of AML cells, and leads to increased DNA damage. In addition to its role in H3K36 tri-methylation, SETD2 is required to maintain high H3K79 di-methylation and MLL-AF9-binding to critical target genes, such as *Hoxa9*. SETD2 loss synergizes with pharmacologic inhibition of the H3K79 methyltransferase DOT1L to induce DNA damage, growth arrest, differentiation, and apoptosis. These results uncover a dependency for SETD2 during MLL-leukemogenesis, revealing a novel actionable vulnerability in this disease.

[1] CeMM Research Center for Molecular Medicine of the Austrian Academy of Sciences, Vienna 1090, Austria. [2] Ludwig Boltzmann Institute for Cancer Research, Vienna 1090, Austria. [3] Research Institute of Molecular Pathology, Vienna 1030, Austria. [4] Cold Spring Harbor Larboratory, Cold Spring Harbor, 11724 NY, USA. [5] MTA TTK Lendület Cancer Biomarker Research Group, Institute of Enzymology, Second Department of Pediatrics, Semmelweis University, Budapest 1094, Hungary. [6] Department of Internal Medicine I. Division of Hematology and Hemostaseology, Ludwig Boltzmann Cluster Oncology, Medical University of Vienna, Vienna 1090, Austria. [7] Center for Physiology and Pharmacology, Medical University of Vienna, Vienna 1090, Austria. [8] Institute for Medical Biochemistry, University of Veterinary Medicine, Vienna 1210, Austria. These authors contributed equally: Johannes Zuber, Giulio Superti-Furga, Florian Grebien.  Correspondence and requests for materials should be addressed to F.G. (email: florian.grebien@vetmeduni.ac.at)

Leukemia-associated fusion proteins serve as a paradigm for modern cancer research, as the molecular machineries that provide their functional cellular context have emerged as amenable to targeted molecular approaches[1,2]. Families of related leukemia fusion proteins that share genomic and biological properties represent unique opportunities to study how the combination of distinct functional protein modules can drive oncogenic transformation. The largest family of "multi-partner translocations" in acute leukemia comprises fusions involving the product of the *KMT2A* (*MLL*) gene. MLL-fusion proteins are found in acute lymphoblastic leukemia (ALL) and acute myeloid leukemia (AML) and are often associated with adverse prognosis, particularly in pediatric patients[3]. Expression of MLL-fusions enhances proliferation and blocks myeloid differentiation of hematopoietic progenitor cells, leading to their pathological accumulation. In line, many MLL-fusions can act as potent oncogenes in cell line models and animal models of leukemia[4].

In leukemia, the MLL N-terminus takes part in >120 different translocations, resulting in the generation of MLL-fusion proteins encompassing more than 75 different partner genes[5]. It has therefore been proposed that the oncogenic activity of MLL-fusion proteins depends on chromatin targeting functions exerted by the MLL N-terminus in combination with other functional properties encoded by the fusion partners[6]. Several regions in the MLL N-terminus are critical for the activity of MLL-fusions. For instance, the CxxC-domain is essential for DNA binding of MLL-fusion proteins[7]. Furthermore, the MLL-interacting protein Menin links MLL-fusion proteins with LEDGF, and the H3K36me3-binding PWWP domain of LEDGF is critical for the function of MLL-fusions[8]. In fact, a direct fusion of the LEDGF PWWP domain to MLL was able to replace Menin altogether[9].

Numerous studies have established strong links between the molecular function of the fusion partner and the mechanistic basis of oncogenic transformation in MLL-fusion-induced leukemogenesis[4]. Pioneering biochemical experiments have shown that several fusion partners of MLL, such as AF4, AF9, and ENL are members of the DOT1L complex (DotCom) and the super-elongation complex (SEC)[10-13], which are both involved in transcriptional control. As the SEC can regulate the transcriptional activity of RNA polymerase II, it was hypothesized that these MLL-fusions induce aberrant regulation of transcriptional elongation on MLL-target genes[14].

A large number of factors was shown to influence the oncogenic properties of MLL-fusions, including signaling proteins[15-17], epigenetic modulators[18-21], and transcription factors[22-24], as well as the wild-type MLL protein[25]. However, it is unclear whether these molecular mechanisms pertain to the entire family of MLL-fusions or if they specifically affect the leukemogenicity of isolated MLL-fusion proteins. In fact, specific molecular mechanisms of oncogenic transformation were postulated to prevail for selected MLL-fusions. For instance, inhibition of the arginine methyltransferase PRMT1 was shown to reduce the leukemic potential of several oncogenic fusion proteins, including MLL-EEN and MLL-GAS7, but not MLL-AF9, MLL-AF10, or MLL-ENL[26,27]. Furthermore, the enzymatic activity of CBP was shown to be required for leukemogenic activity of fusions of MLL with the histone acetyltransferase CREBBP[28,29]. Finally, dimerization might play an important role in nuclear translocation and oncogenic transformation in fusions of MLL to the cytoplasmic partner proteins GAS7 and AF1p, yet the underlying molecular mechanism is unclear[30,31].

Here, we set out to survey the molecular composition of a diverse subset of distantly related MLL-fusion protein complexes to characterize their unique and common properties, and to reveal possible actionable vulnerabilities that are based on specific molecular mechanisms shared by MLL-fusions. We identify the methyltransferase SETD2 as an interactor of all MLL-fusion proteins. shRNA-mediated and CRISPR/Cas9-mediated loss of SETD2 leads to growth arrest and differentiation of MLL-fusion-expressing cells in vitro and in vivo. Moreover, we show that loss of SETD2 is associated with increased DNA damage. SETD2 loss disrupts a H3K36me3-H3K79me2 signature on MLL-target genes and sensitizes MLL-AML cells to pharmacologic inhibition of the known MLL-fusion protein effector DOT1L. In summary, we describe a novel dependency for SETD2 in the initiation and maintenance of MLL-rearranged leukemia, highlighting a novel vulnerability in this disease.

## Results

**Functional proteomic survey of MLL-fusion proteins.** Reasoning that critical effectors might be enriched among the physical interaction partners of distantly related MLL-fusion proteins, we undertook an unbiased survey of the protein–protein interactions of MLL-fusion proteins in leukemia cells. Using FRT/Flp-mediated locus-specific cassette exchange, we generated isogenic Jurkat leukemia cell lines allowing for Doxycycline (Dox)-inducible, single-copy expression of affinity-tagged variants of seven MLL-fusions that were previously proposed to employ different molecular mechanisms of oncogenic transformation (MLL-AF1p, MLL-AF4, MLL-AF9, MLL-CBP, MLL-EEN, MLL-ENL, MLL-GAS7, Fig. 1a, b and Supplementary Fig. 1a-c). Subcellular fractionation revealed that all selected MLL-fusion proteins localized to the nucleus (Supplementary Fig. 1d) and were capable of inducing expression of the MLL-fusion-target genes *HOXA5*, *HOXA9*, *HOXA10*, and *MEIS1* (Fig. 1c).

Protein complexes around MLL-fusion proteins were purified from nuclear lysates of cell lines expressing seven distinct MLL-fusions (Fig. 1d and Supplementary Fig. 2a). Purifications were analyzed by LC-MS/MS using both one-dimensional and two-dimensional gel-free proteomic approaches, recovering 4600 proteins in total, engaging in 15,094 putative interactions (Fig. 1e and Supplementary Fig. 2b)[32-34]. *p*-value-based filtering for the 300 most significant interactions per MLL-fusion resulted in a network of 960 high-confidence cellular proteins (Supplementary Fig. 2b). Validation of the network confirmed previously reported interactions of MLL-fusions with protein complexes important for transcriptional control and epigenetic regulation, including the PAF complex, the SWI/SNF complex, and Polycomb Repressor Complex 1 (Supplementary Fig. 2c)[12,35-37]. The network also revealed abundant unique interaction partners of all MLL-fusion proteins, indicating that distinct MLL-fusions can engage specific molecular pathways. 406 proteins in the network (42.3%) co-purified with more than one MLL-fusion protein, while 128 proteins (13.3%) interacted with at least five of the seven MLL-fusions (Supplementary Fig. 2b), indicating a strong degree of topological conservation within the MLL-interaction network. Further analysis of the 128 conserved partners of MLL-fusion proteins revealed six distinct protein communities (p < 0.01; Supplementary Table 1), whose annotation retrieved molecular functions that are highly relevant to the biology of MLL-fusion proteins, including chromatin remodeling, transcriptional elongation, and hematopoiesis (Fig. 1e). Interestingly, protein families that had not been reported to interact with MLL-fusion proteins before were also identified, such as factors involved in DNA-repair, RNA splicing, and RNA transport.

In summary, our comprehensive analysis and validation of the cellular interaction networks shows that distinct MLL-fusion proteins engage in a high number of direct, as well as indirect protein–protein interactions. Structurally different MLL-fusion proteins share 128 conserved interaction partners, which are

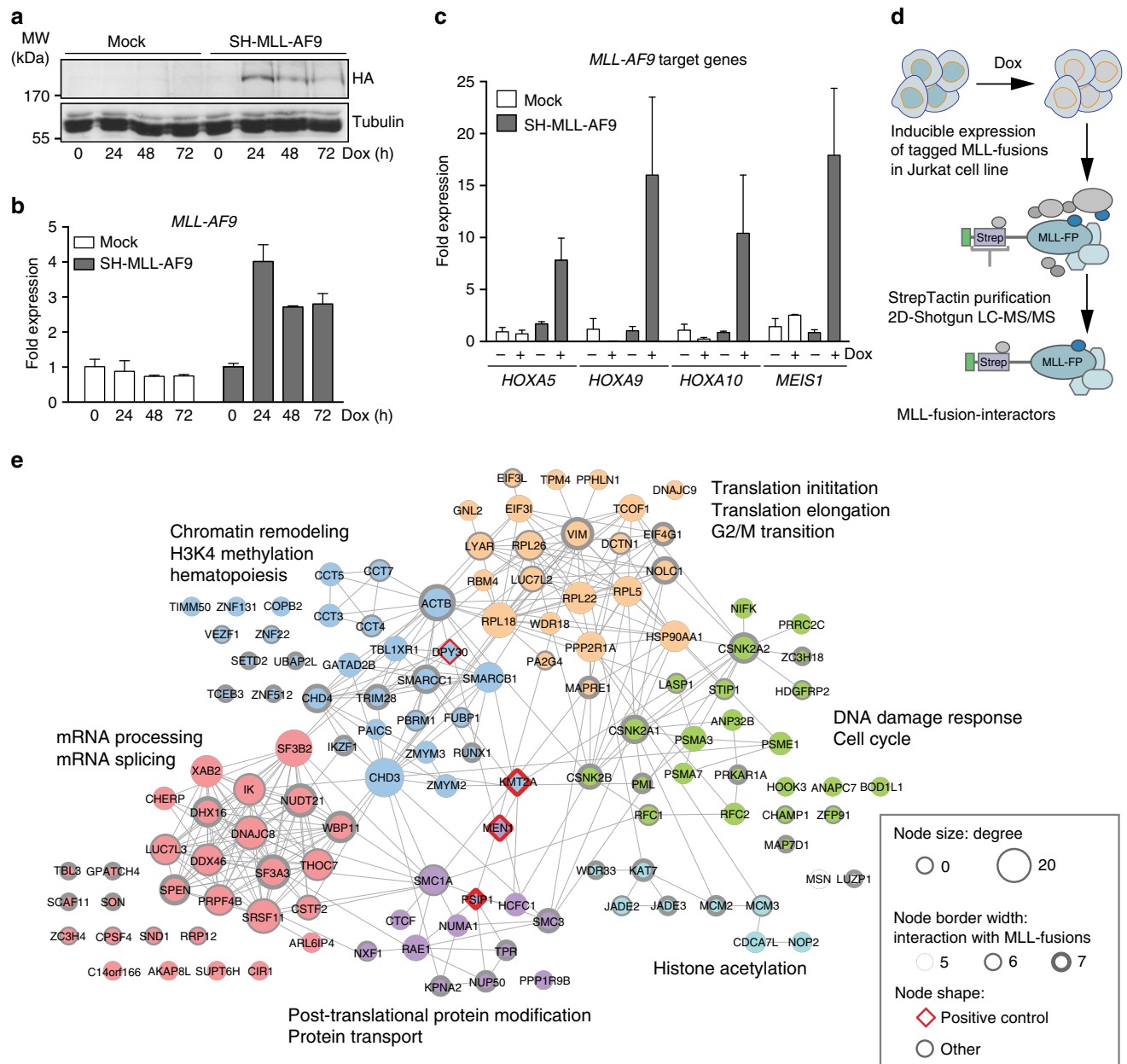

**Fig. 1** Functional proteomic survey of the MLL-fusion interactome. Cells expressing Strep-HA (SH)-tagged MLL-AF9 or mock-transfected cells were treated with Dox for the indicated time points and transgene expression was monitored by immunoblotting (**a**) and qPCR (**b**) (means ± s.d. n = 3). **c** SH-MLL-AF9-expressing cells were treated with Dox for 24 h and the expression of indicated MLL-target genes was measured by qPCR (mean ± s.d. n = 3). **d** Schematic illustration of the strategy of affinity purification of protein complexes associated with MLL-fusion proteins from nuclear lysates of cell lines expressing affinity-tagged MLL-fusion proteins. **e** Gene ontology (GO) enrichment of six distinct protein communities among the core 128 interactors shared by at least 5 of 7 MLL-fusion proteins

enriched in six functional communities that are highly relevant for the biology of MLL-fusion proteins.

**shRNA screen identifies SETD2 as an effector of MLL-fusions**. As our primary validation reduced the number of potential critical effectors in the network of MLL-fusion protein-interactors from 960 to 128, we next aimed to further narrow down the circle of candidate proteins using sequential functional genomic approaches (Fig. 2a). To systematically investigate the functional contribution of the conserved 128 MLL-interaction partners to MLL-fusion-dependent leukemia, we devised a shRNA screen in the human MLL-AF9-expressing AML cell line MOLM-13. In the

system used by us, transcriptional coupling of fluorescent reporter proteins to shRNA expression upon Dox-induction allows for dynamic monitoring of competing growth kinetics in mixed cell populations expressing experimental shRNAs (GFP) vs. non-targeting control shRNAs (shRen.713, dsRed, Fig. 2b). While expression of a control shRNA did not differentially affect cell proliferation in mixed populations over time, strong shRNA-induced negative selection of GFP-positive cells was observed upon targeting of *MEN1*, an interaction partner of all seven investigated MLL-fusion proteins with a well-known function in MLL-fusion-induced leukemogenesis[38] (Fig. 2b, bottom). We used this setup to systematically test the effects of 128 shRNA-pools targeting conserved MLL-fusion interaction partners on

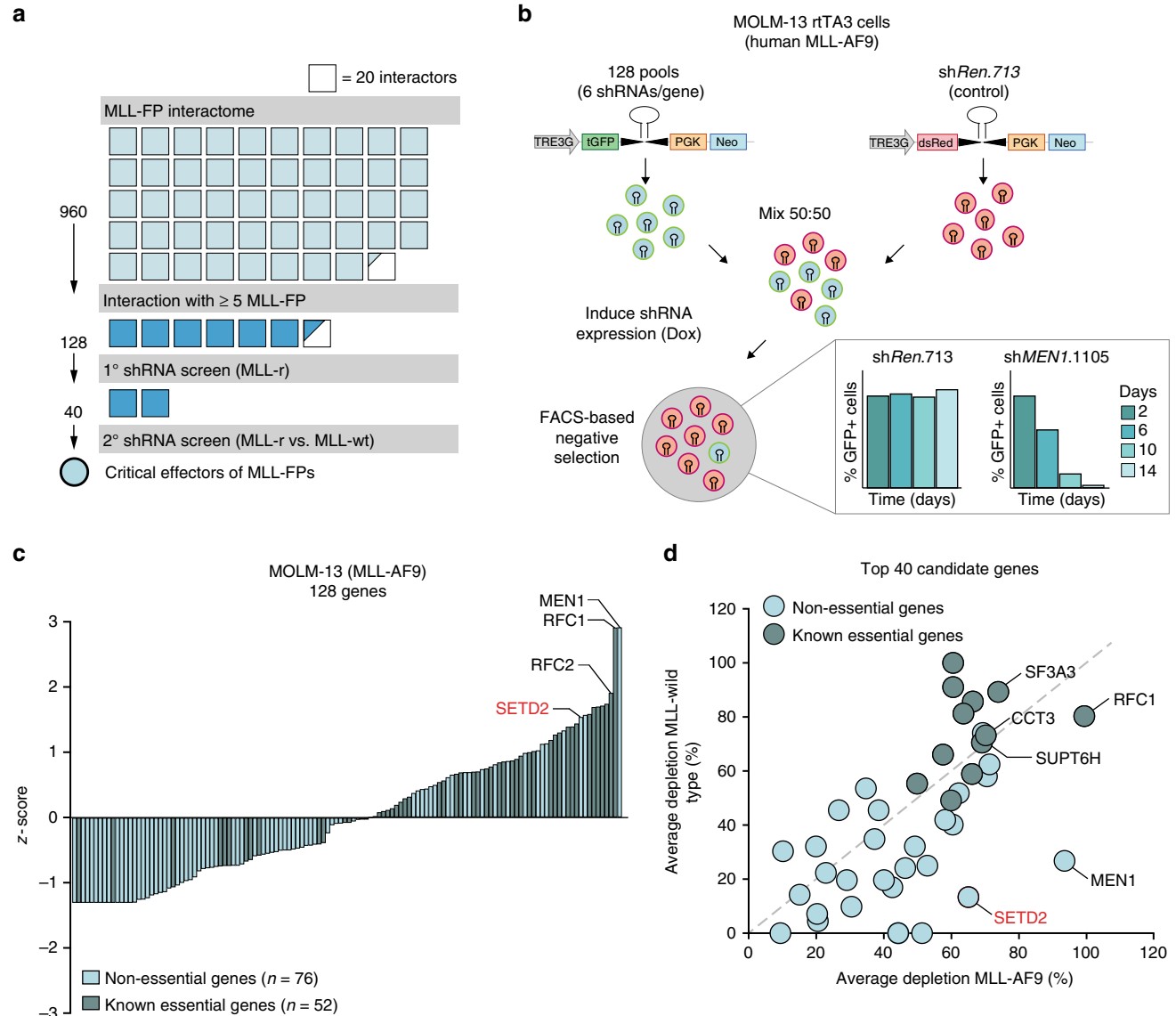

**Fig. 2** shRNA screen identifies SETD2 as a critical effector of MLL-fusions. **a** Schematic representation of the filtering strategy. Affinity purification coupled to mass spectrometry identified 960 candidate genes (top 300 interactors per bait, ranked by *p*-value) to interact with at least one of seven selected MLL-fusion proteins. 128 proteins interacted with ≥5 of seven MLL-fusions. 40 candidate genes were screened in MLL-rearranged vs. MLL-wild-type cell lines. Each square corresponds to 20 interactors. **b** Schematic outline of retroviral vectors and experimental design of the FACS-based negative selection RNAi screen. Competitive proliferation assays were set up by mixing cells in a 50:50 ratio (experimental-GFP vs. control-dsRed) and cultivation in the presence of Dox. The relative ratio of GFP-positive vs. dsRed-positive cells was monitored by flow cytometry over 14 days. Bar graphs (bottom) represent the performance of positive (sh*MEN1*.1105) and negative (sh*Ren*.713) controls shown as percentage of GFP+ cells over time. **c** Summary of RNAi screening data in the MOLM-13 cell line. Positive and negative *z*-scores correspond to candidate genes with stronger and weaker depletion phenotypes. Gene essentiality was assigned based on published datasets. **d** Average depletion values obtained from two subsequent RNAi screens performed in the MLL-AF9-expressing MOLM-13 cell line are plotted against mean depletion values from counterscreens performed in two MLL-wild-type cell lines (K562 and HL-60)

AML cell growth. Relative depletion of all shRNA-pools was normalized to a negative-control shRNA (sh*Ren*.713) and to a strong growth inhibitory positive-control shRNA (sh*Myb*.670)[22]. As the read-out of this screen is inhibition of proliferation, we would expect that essential genes would be enriched among the strongest hits. Indeed, scoring of shRNA-induced effects upon knockdown of all 128 MLL-fusion interactors revealed a strong positive correlation between growth inhibition and reported gene essentiality (Fig. 2c)[39–42]. However, as we intended to identify proteins with MLL-fusion-specific roles in the network, we reasoned that their loss-of-function might preferably affect the viability of MLL-fusion-expressing leukemia cells. Thus, we re-

screened the 40 candidate genes with the highest confidence in MLL-AF9-expressing MOLM-13 cells and in the MLL-wild-type leukemia cell lines K562 and HL-60. As expected, knockdown of MLL-interactors with known essential functions, such as *RFC1*, *SF3A3*, or *CCT3*, led to growth inhibition in both MLL-fusion cells and MLL-wild-type leukemia cells (Fig. 2d). In contrast, depletion of the known MLL-interaction partner *MEN1* selectively inhibited growth of MLL-fusion cells while sparing MLL-wild-type cells, proving the validity of the chosen strategy.

Knockdown of the gene encoding the H3K36me3-specific methyltransferase *SETD2* showed a strong bias towards inhibition of proliferation of MLL-fusion-expressing cells, while causing

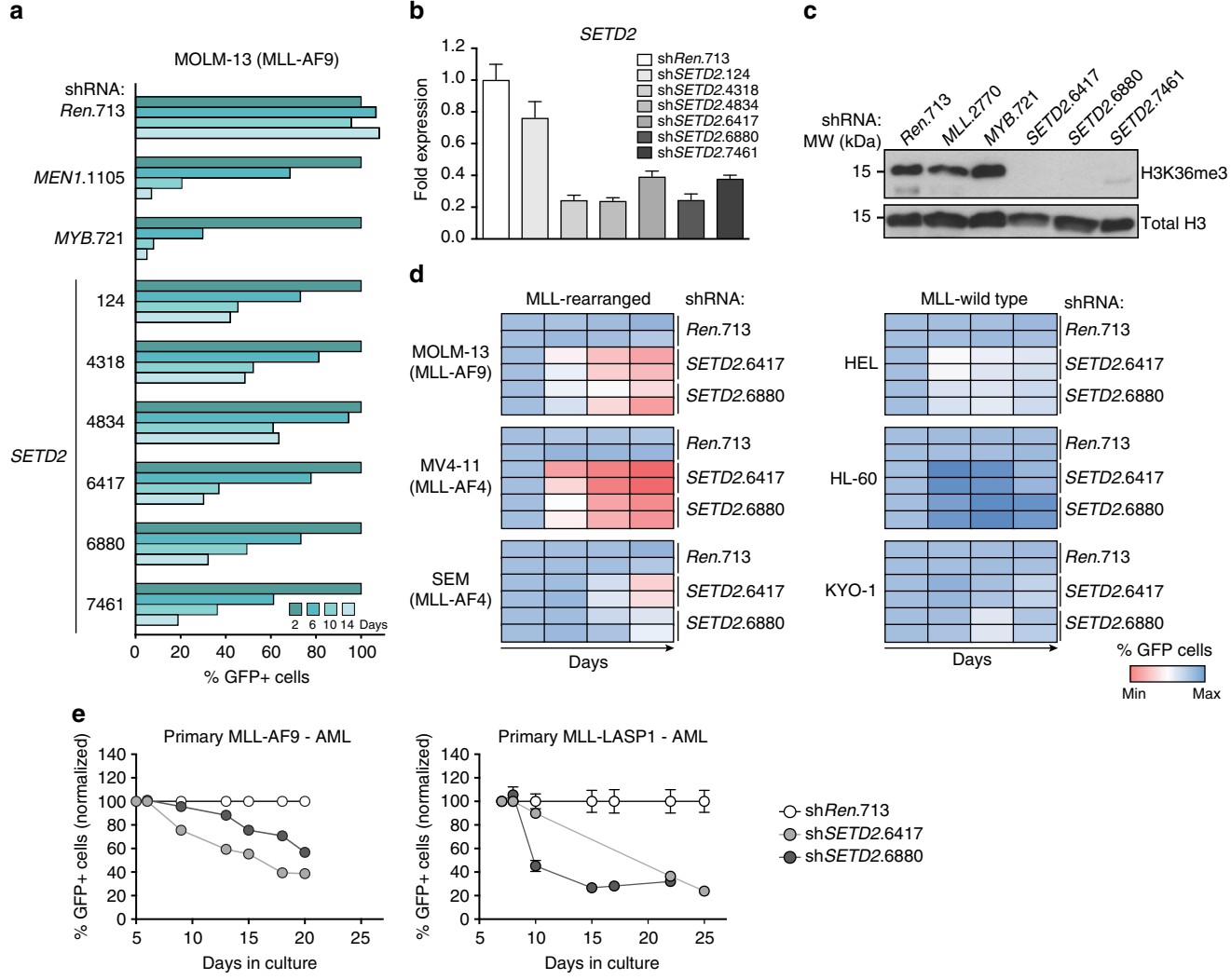

**Fig. 3** SETD2 is required for proliferation of MLL-leukemia cells. **a** Results of FACS-based competitive proliferation assay shown as the percentage of GFP-positive MOLM-13 cells expressing individual *SETD2*-targeting shRNAs in the presence of Dox over 14 days. One representative experiment of four is shown. **b** qPCR analysis of *SETD2* mRNA levels in MOLM-13 cells expressing indicated shRNAs after 48 h of Dox treatment (mean ± s.d. n = 3). **c** Western blot analysis of H3K36me3 levels in MOLM-13 cells expressing indicated shRNAs after 72 h of Dox treatment. **d** Heatmap representation of competitive proliferation assays performed in human cell lines harboring MLL rearrangements (left) vs. MLL-wild-type cells (right) expressing indicated shRNAs targeting *SETD2* as described in **a**. Representative results of two out of three experiments are shown. **e** Time course of GFP expression of primary human AML cells from patients expressing *MLL-AF9* and *MLL-LASP1* fusion genes expressing indicated shRNAs (mean ± s.d. n = 3)

negligible cell depletion in K562 and HL-60 cells, suggesting an MLL-fusion-specific function (Fig. 2d, Supplementary Fig. 3a). SETD2 was one of 42 core proteins that interacted with all seven MLL-fusions, as it co-purified with MLL-fusion proteins in all affinity-purification experiments with significant peptide coverage (Supplementary Fig. 3b). Consistently, co-immunoprecipitation experiments showed that this interaction involved the N-terminal part of MLL, which is conserved in all MLL-fusion proteins studied, and the C-terminus of SETD2, which encompasses all annotated functional domains of the SETD2 protein (Supplementary Fig. 3c). *SETD2* expression was higher in AML samples than in normal hematopoietic stem and progenitor cell types and mature myeloid cells[43] (Supplementary Fig. 3d). *SETD2* expression was highest in patients with 11q23 aberrations featuring MLL-translocations, as compared to samples with normal karyotype AML or myelodysplastic syndrome (Supplementary Fig. 3e).

Thus, we identified the methyltransferase *SETD2* as a selective effector of MLL-AF9 AML cells through functional genomic investigation of conserved interaction partners of MLL-fusion proteins.

**SETD2 is essential for MLL-fusion-expressing cells**. We next aimed at validating the shRNA screen results at the level of individual shRNAs. Expression of all six *SETD2*-targeting shRNAs induced strong growth inhibition in MOLM-13 cells, in line with significant downregulation of *SETD2* mRNA (Fig. 3a, b). As *SETD2* is the only protein known to mediate tri-methylation of H3K36[44], we investigated the effect of *SETD2* downregulation on total cellular H3K36me3 levels. The three strongest *SETD2*-targeting shRNAs caused near-complete clearance of global H3K36me3 signals (Fig. 3c). Importantly, growth inhibition was not generally associated with H3K36me3 loss, as we did not observe changes in global H3K36me3 levels upon downregulation of *MLL* and *MYB*, which strongly affected proliferation of MLL-AF9 AML cells (Fig. 3c).

As our screening data indicate that *SETD2* knockdown selectively inhibits the proliferation of MLL-fusion-expressing cells, we sought to extend this observation to a larger panel of human leukemia cell lines. In addition to MOLM-13 cells also the MLL-AF4-expressing cell lines MV4-11 and SEM showed

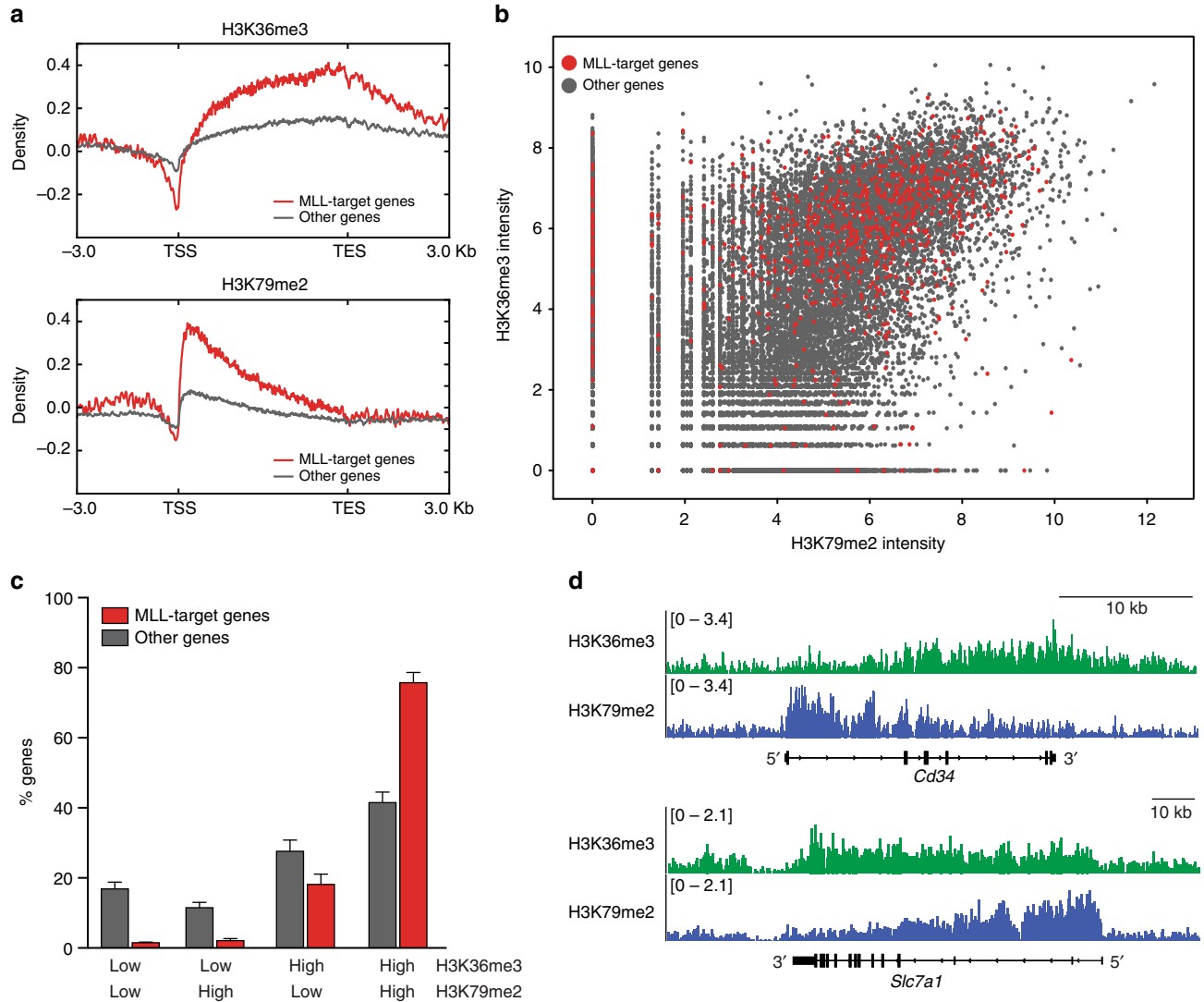

**Fig. 4** MLL-target genes are marked by a H3K36me3-H3K79me2 signature. **a** Metagene plots of ChIP-Rx data for H3K36me3 (top) and H3K79me2 (bottom) for MLL-target genes (red) or non-MLL-target genes (gray) in mouse *MLL-AF9/Nras*G12D AML cells. **b** Dot plot of normalized ChIP-Rx signal intensities for H3K36me3 vs. H3K79me2 marks on MLL-target genes (red) or non-MLL-target genes (gray) in the mouse genome (Pearson $R = 0.44$, $p < 2.2 \times 10^{-16}$). **c** Bar graph showing percentages of genes among MLL-target genes and non-MLL-target genes associated with the indicated histone marks; low: not exceeding input counts per gene, high: exceeding input counts per gene (mean ± s.d. $n = 2$). **d** H3K36me3 (green) and H3K79me2 profiles (blue) of selected MLL-AF9-target genes

significant anti-proliferative responses and induction of apoptosis upon *SETD2* knockdown (Fig. 3d, left, and Supplementary Fig. 4a-c). In contrast, *SETD2* downregulation in the MLL-wild-type cell lines HEL, HL-60, and KYO-1 only marginally affected proliferation (Fig. 3d, right, Supplementary Fig. 4c). *SETD2* knockdown resulted in a strong proliferative disadvantage in primary human AML cells from patients expressing *MLL-AF9* and *MLL-LASP1* fusion genes (Fig. 3e).

Taken together, downregulation of *SETD2* caused a strong anti-proliferative response in primary human AML cells and cell lines expressing various MLL-fusion genes, suggesting a requirement for *SETD2* in the oncogenic context of MLL-fusion proteins.

**MLL-target genes exhibit high H3K36me3 levels**. To investigate the relationship between SETD2 and MLL-fusions we profiled the global distribution of the SETD2-dependent H3K36me3 mark in a

mouse AML cell line expressing *MLL-AF9* and activated *Nras* (G12D)[22] using ChIP-Rx[45]. As expected, H3K36me3 was present on gene bodies of expressed genes. We found that MLL-AF9 target genes[22] displayed significantly higher H3K36me3 levels than non-MLL-target genes (Fig. 4a, top). In line with previous data, MLL-fusion target genes were also highly positive for the DOT1L-dependent H3K79me2 mark[21] (Fig. 4a, bottom), and the global levels of H3K36me3 and H3K79me2 modifications showed a strong positive correlation in mouse *MLL-AF9/Nras*G12D cells (Fig. 4b). However, while only 42% of non-MLL-target genes were highly positive for both marks, 76% of MLL-target genes displayed a combined H3K36me3/H3K79me2-high signature (Fig. 4c, d, and Supplementary Fig. 5). As MLL-fusion-binding was shown to correlate with H3K79me2 on MLL-target genes[21] and depend on recognition of H3K36me3 marks[9], these data suggest that the SETD2-dependent H3K36me3 modification is part of an epigenetic signature that marks target genes of MLL-fusion proteins together with H3K79me2.

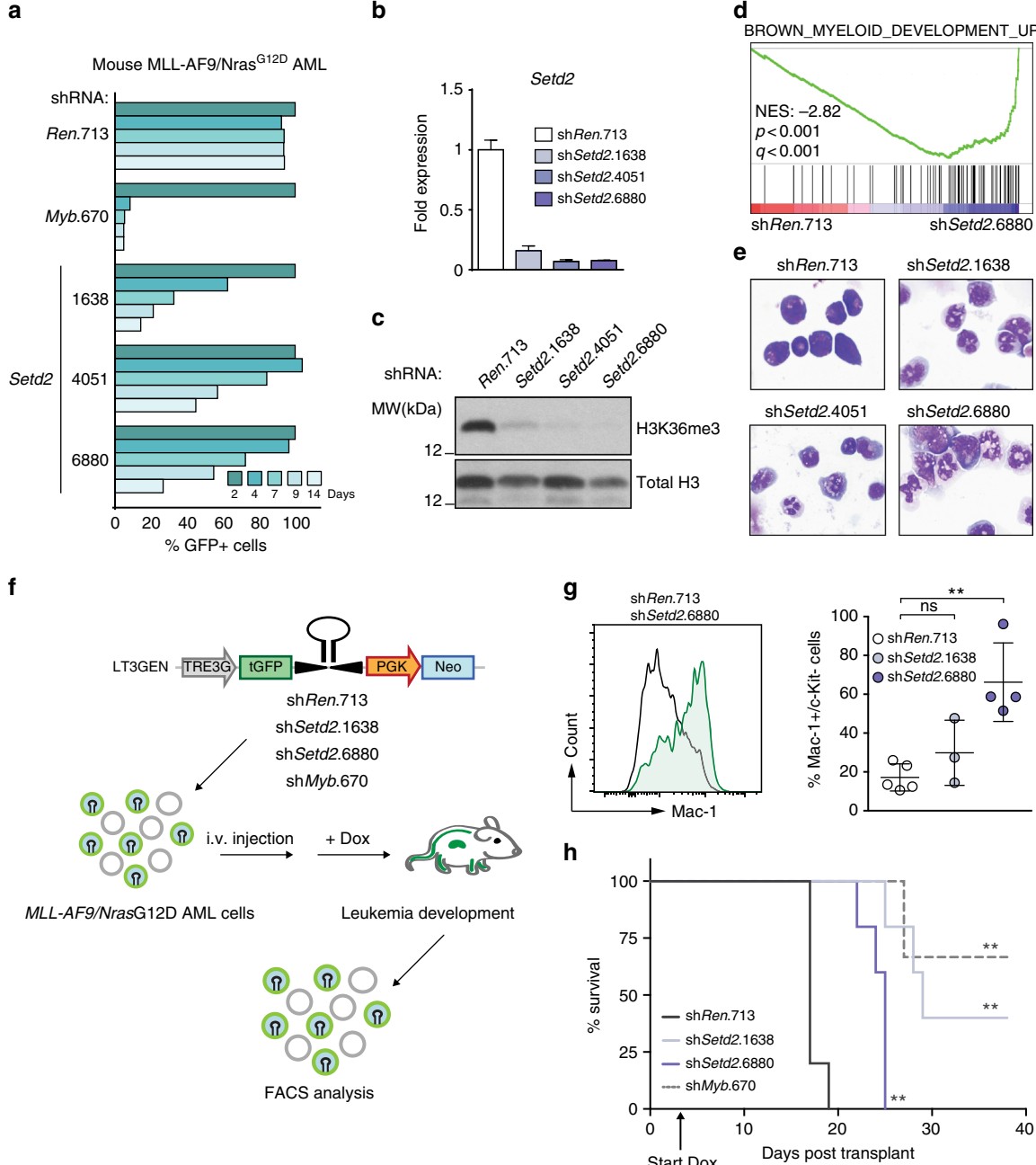

**Fig. 5** SETD2 loss induces myeloid differentiation of MLL-leukemia cells. **a** Results of competitive proliferation assay shown as the percentage of GFP-positive mouse *MLL-AF9/Nras*G12D AML cells expressing indicated shRNAs in the presence of Dox over 14 days. One representative experiment out of three is shown. **b** qPCR analysis of *Setd2* mRNA levels in *MLL-AF9/Nras*G12D cells expressing indicated shRNAs after 48 h of Dox treatment (mean ± s.d. *n* = 3). **c** Western blot analysis of H3K36me3 levels in *MLL-AF9/Nras*G12D cells expressing indicated shRNAs after 72 h of Dox treatment. **d** Gene Set Enrichment Analysis indicating myeloid differentiation of *MLL-AF9/Nras*G12D AML cell line upon knockdown of *Setd2*. NES, Normalized Enrichment Score. **e** Micrographs of cytospin preparations of *MLL-AF9/Nras*G12D AML cells after expression of indicated shRNAs. **f** Schematic representation of the in vivo transplantation assay. *MLL-AF9/Nras*G12D AML cells expressing indicated shRNAs were transplanted into sub-lethally irradiated C57BL/6 Ly5.1 mice. Dox was administrated to the drinking water starting from day 3 (arrow in **h**) and disease progression was monitored by bioluminescence imaging. Terminally sick mice were sacrificed and analyzed. **g** Flow cytometric analysis of Mac-1 on *MLL-AF9/Nras*G12D AML cells upon shRNA-mediated *Setd2* knockdown in vivo (mean ± s.d. *n* ≥ 3). **h** Kaplan–Meier survival curves of C57BL/6 Ly5.1 mice transplanted with *MLL-AF9/Nras*G12D AML cells expressing *Setd2*-targeting shRNAs. Survival curves of mice transplanted with cells expressing *Setd2*-targeting shRNAs were compared to cells expressing control shRNAs using a Log-rank test. ns, not significant, **$p < 0.01$ (*t*-test)

**Loss of SETD2 induces myeloid differentiation and DNA damage.** Next we sought to characterize global changes in gene expression upon SETD2 ablation. Dox-inducible knockdown of *Setd2* caused a strong growth disadvantage in mouse *MLL-AF9/*

*Nras*G12D cells (Fig. 5a, b, Supplementary Fig. 6a). *Setd2* downregulation led to almost complete loss of cellular H3K36me3 signals (Fig. 5c). RNA-seq analysis showed that 868 genes were differentially expressed upon *Setd2* knockdown in

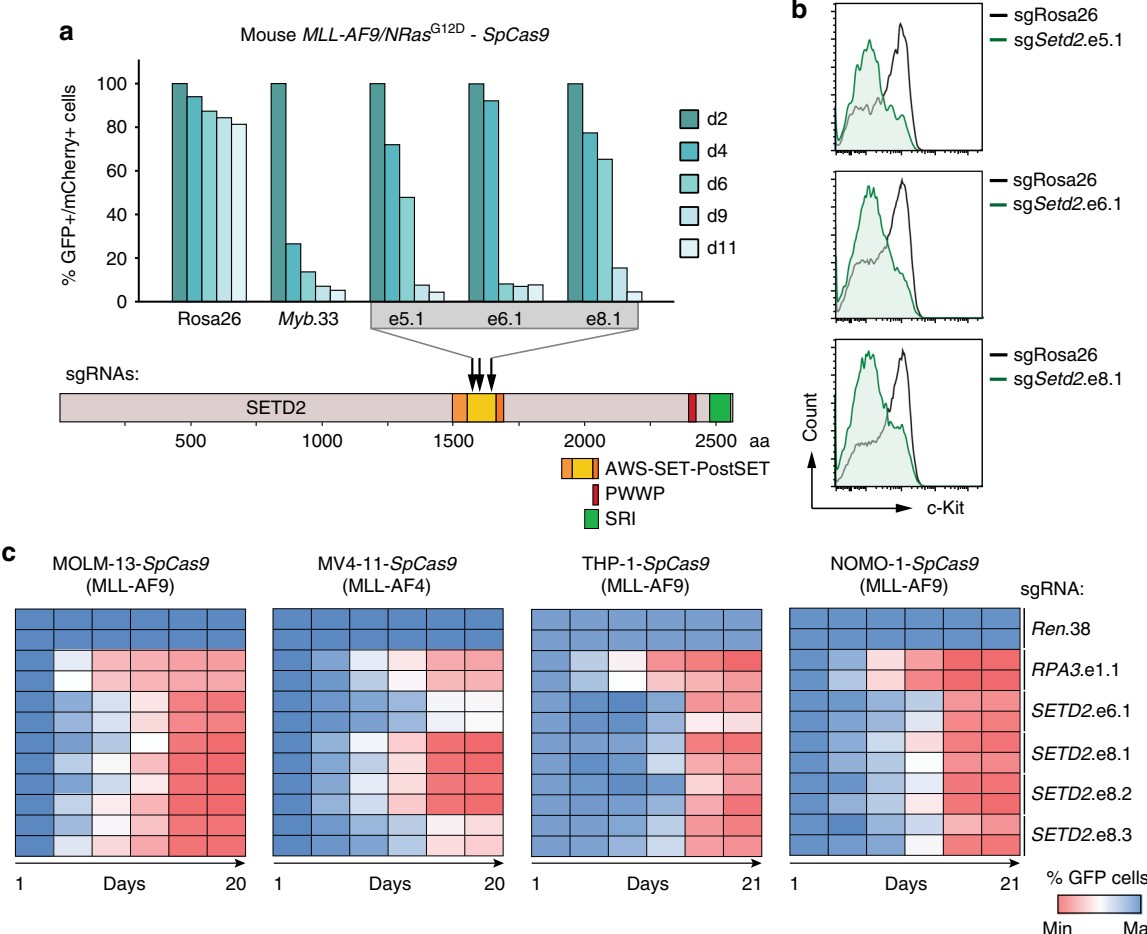

**Fig. 6** The SETD2 SET domain is required for proliferation of MLL-leukemia cells. **a** Results of competitive proliferation assays shown as percentages of mCherry+/GFP+ mouse *MLL-AF9/Nras*G12D-SpCas9 AML cells expressing indicated sgRNAs (top). Data from one representative experiment of two are shown. Schematic representation of the domain structure of *SETD2* (bottom). **b** Flow cytometric analysis of surface expression of c-Kit in *MLL-AF9/ Nras*G12D AML-SpCas9 cells upon CRISPR/Cas9-mediated mutagenesis of *SETD2*. **c** Heatmap representation of competitive proliferation assays shown as the percentage of GFP-positive human AML cell lines stably expressing *SpCas9* cells upon CRISPR/Cas9-mediated mutagenesis of the *SETD2* SET domain

*MLL-AF9/Nras*G12D AML cells. While 458 genes were upregulated, 410 genes were downregulated in sh*Setd2*-cells, (padj < 0.01, Supplementary Fig. 6b). Consistent with a role of SETD2 in the DNA damage response[46], *MLL-AF9/Nras*G12D AML cells expressing two different *Setd2*-targeting shRNAs showed upregulation of DNA damage-associated gene expression (Supplementary Fig. 6c). Indeed, *Setd2* downregulation resulted in significantly higher levels of DNA damage in the absence of genotoxic agents, as measured by alkaline comet assay and phosphorylated histone H2AX (γ-H2AX, Supplementary Fig. 7a-c). Knockdown of *Setd2* led to induction of p21, reduced cell cycle progression, and induction of apoptosis of *MLL-AF9/Nras*G12D AML cells (Supplementary Fig 7c-e).

Gene Set Enrichment Analysis revealed that *Setd2* downregulation induced gene expression programs associated with myeloid differentiation (Fig. 5d). Indeed, *Setd2*-deficient cells displayed clear signs of terminal myeloid maturation, including nuclear segmentation and increased granularity (Fig. 5e), as well as downregulation of the progenitor marker c-Kit and upregulation of the mature myeloid marker Mac-1 (Supplementary Fig. 8a). Similarly, *SETD2* downregulation in the human MLL-AF4-expressing cell line MV4-11 and in MLL-AF9-expressing MOLM-13 cells induced increased cell surface levels of the differentiation marker CD36 together with macroscopic changes characteristic of myeloid maturation (Supplementary Fig. 8b, c).

To test whether loss of SETD2 could overcome the MLL-AF9-dependent differentiation block in myeloid progenitors in vivo, we transplanted *MLL-AF9/Nras*G12D AML cells expressing *Setd2*-targeting or control shRNAs into recipient mice. shRNA expression was induced by Dox-administration and the immuno-phenotype of the developing leukemia was analyzed by flow cytometry (Fig. 5f). Knockdown of *Setd2* induced strong downregulation of c-Kit concomitant with upregulation of Mac-1 in leukemic cells in vivo, resulting in a significant increase in disease latency (Fig. 5g, h and Supplementary Fig. 8d). This is consistent with recent results showing that knockout of *Setd2* greatly increased the latency of MLL-AF9-induced AML[47]. While most leukemia cells isolated from moribund recipients of control AML cells showed robust shRNA expression (as measured by GFP levels), shRNA-expressing cells were strongly outcompeted by shRNA-negative cells in case of *Setd2* knockdown in vivo (Supplementary Fig. 8e).

In summary, shRNA-mediated downregulation of SETD2 caused growth arrest, induction of apoptosis, and increased DNA damage. Furthermore, SETD2 loss induced terminal myeloid differentiation in MLL-fusion-expressing mouse and human AML cells in vitro and in vivo, indicating that the MLL-fusion-induced differentiation block is SETD2-dependent.

**The SETD2 SET domain is required for AML growth**. To interrogate the translational potential of our findings, we next

wanted to establish whether the methyltransferase activity of SETD2 is necessary for the observed effects. Direction of SpCas9-cleavage to functional protein domains was shown to greatly increase the read-out in competitive proliferation assays[48]. We employed CRISPR/Cas9-mediated mutagenesis of the enzymatic SET domain to investigate whether catalytic activity of SETD2 was required for the oncogenicity of MLL-fusion proteins. Introduction of three sgRNAs targeting the Setd2 SET domain in SpCas9-expressing MLL-AF9/NrasG12D AML cells led to a strong depletion of transduced cells over time, as shown before[48] (Fig. 6a). Notably, CRISPR/Cas9-mediated mutagenesis of the Setd2 SET domain was sufficient to induce myeloid differentiation of MLL-AF9/NrasG12D cells, as measured by down-regulation of c-Kit together with upregulation of Mac-1 (Fig. 6b, Supplementary Fig. 9a). In line, we found strong anti-proliferative effects, induction of myeloid differentiation, and apoptosis upon mutagenesis of the SETD2 SET domain in the human MLL-rearranged AML cell lines MOLM-13 and MV4-11, THP-1, and NOMO-1 (Fig. 6c; Supplementary Fig. 9b-f, Supplementary Fig. 10).

These data show that the SET domain of SETD2 is required to sustain the proliferative capacity and differentiation block of MLL-fusion protein-expressing AML cells. In addition, these results imply a functional involvement of the H3K36me3 mark in the maintenance of MLL-fusion-dependent leukemia and offer a plausible route for future pharmacological intervention.

**Efficient MLL-fusion-mediated transformation requires SETD2**. All our data show a strong functional requirement for the expression and activity of SETD2 in the progression of MLL-leukemia. As it is possible that alternative molecular mechanisms pertain during initiation of MLL-rearranged leukemia, we tested the involvement of SETD2 in this process. Setd2 knockdown resulted in a significant reduction in MLL-AF9-induced serial re-plating capacity of mouse hematopoietic stem/progenitor cells (HSPC), indicating that Setd2 expression is required to unleash the full oncogenic potential of MLL-AF9 (Fig. 7a). Setd2 ablation induced loss of compact colony morphology characteristic of blast-like cells and induced the formation of large, dispersed colonies reminiscent of mature myeloid clusters (Fig. 7b). Flow cytometry confirmed that Setd2-deficient colonies expressed high levels of the mature myeloid marker Mac-1 (Fig. 7c). To investigate the effect of SETD2 on oncogenic transformation in vivo, we co-transduced fetal-liver-derived HSPC expressing a SpCas9 transgene[49] with retroviral vectors encoding MLL-ENL and Setd2-targeting or control sgRNAs. The contribution of cells carrying sgRNA-induced mutations in the Setd2 SET domain to leukemia development was investigated by flow cytometric analysis of mCherry expression upon transplantation (Fig. 7d). While cells expressing a control sgRNA showed robust contribution to MLL-ENL-induced leukemia in vivo (56%), cells carrying Setd2-mutations induced by two different sgRNAs were clearly under-represented in the leukemic population (5–25%, Fig. 7e, f).

Thus, both downregulation and mutagenesis of SETD2 was incompatible with efficient oncogenic transformation by MLL-fusion oncoproteins in vitro and in vivo. These results indicate that SETD2 expression is required for leukemogenesis and establish SETD2 as a novel actionable target in MLL-rearranged leukemia.

**SETD2 loss sensitizes MLL-AML cells to DOT1L inhibition**. Finally, we aimed to obtain insight into the molecular mechanism that functionally connects SETD2 activity with MLL-fusion-induced leukemia. ChIP-Rx showed that Setd2 downregulation led to a concomitant reduction of both H3K36me3 and H3K79me2 levels on MLL-target genes (Fig. 8a, b and Supplementary Fig 11a), while it did not alter H3K4me3 density (Supplementary Fig. 11b). As chromatin binding of MLL-fusion proteins was shown to depend on H3K36me3 recognition via the conserved interaction partner LEDGF[9], we hypothesized that reduction of H3K36me3 levels upon Setd2 loss might impair chromatin binding of MLL-fusions. Indeed, knockdown of Setd2 caused reduced binding of MLL-AF9 to the promoters of the canonical MLL-target genes Hoxa9 and Meis1 (Fig. 8c), leading to reduced Hoxa9 expression (Fig. 8d).

Given the dependence of the dual H3K36me3-H3K79me2 signature across MLL-target genes on SETD2 and the strong requirement of MLL-leukemia for the H3K79 methyltransferase DOT1L[21], we reasoned that SETD2 loss might cooperate with pharmacologic inhibition of DOT1L. Treatment of mouse MLL-AF9/NrasG12D and human MLL-AF4-expressing MV4-11 cells with the clinical DOT1L inhibitor EPZ5676[50] potentiated the effects of SETD2 downregulation, including growth inhibition, induction of apoptosis, and onset of myeloid differentiation. Importantly, none of these parameters were altered in SETD2-proficient cells in the presence of the same concentrations of the inhibitor (Fig. 8e, f, Supplementary Fig. 11c-f). Finally, and consistent with a role of DOT1L in DNA repair[51], we found that combination of SETD2 loss and DOT1L inhibition synergized in the induction of DNA damage (Fig. 8g and Supplementary Fig. 11g).

These data show that loss of SETD2 expression in MLL-fusion AML cells interferes with the H3K36me3-H3K79me2-signature on MLL-target genes and impairs chromatin binding of MLL-fusion proteins. In consequence, SETD2 loss led to hyper-sensitization of MLL-leukemia cells to small-molecule-mediated DOT1L inhibition, which provides a rationale for potential future combination therapies in AML.

**Discussion**

Here, we provide the first comprehensive protein–protein interaction network of MLL-fusion proteins in leukemia cells. We show that functional annotation of conserved MLL-interaction partners by loss-of-function screening enables the identification of conserved actionable nodes among the molecular network of MLL-fusions. As exemplified by our discovery of the histone methyltransferase SETD2 as an essential factor in MLL-rearranged leukemia, this approach can reveal novel genetic dependencies and yield new entry points for targeting of the entire group of MLL-rearranged leukemia, comprising over 75 different MLL-fusion partners.

Our results show that MLL-fusion proteins engage a large number of distinct protein–protein interactions. This could be explained by the modular nature of wild-type core MLL complexes[52,53] and by the specific architecture of their leukemic counterparts[4]. Our analysis of protein–protein interactions of selected, molecularly distinct MLL-fusion proteins greatly expands the cellular catalog of MLL-interacting proteins. In addition, it also provides novel insights into the topologies of MLL-fusion proteins that transform cells via unknown mechanisms. For instance, interactome analysis of the MLL-GAS7-fusion protein showed that it specifically interacts with components of the CTLH complex, which is involved in microtubule dynamics and chromosome segregation[54].

A core set of 128 proteins constitutes the conserved interactome of MLL-fusion proteins. In addition to known interaction partners of the wild-type MLL protein, such as MEN1, DPY30, and LEDGF, it also contains several proteins whose link to AML biology have only recently been established. For instance, the protein SON interacts with MEN1 to regulate the expression of leukemia-specific genes in a MLL-dependent

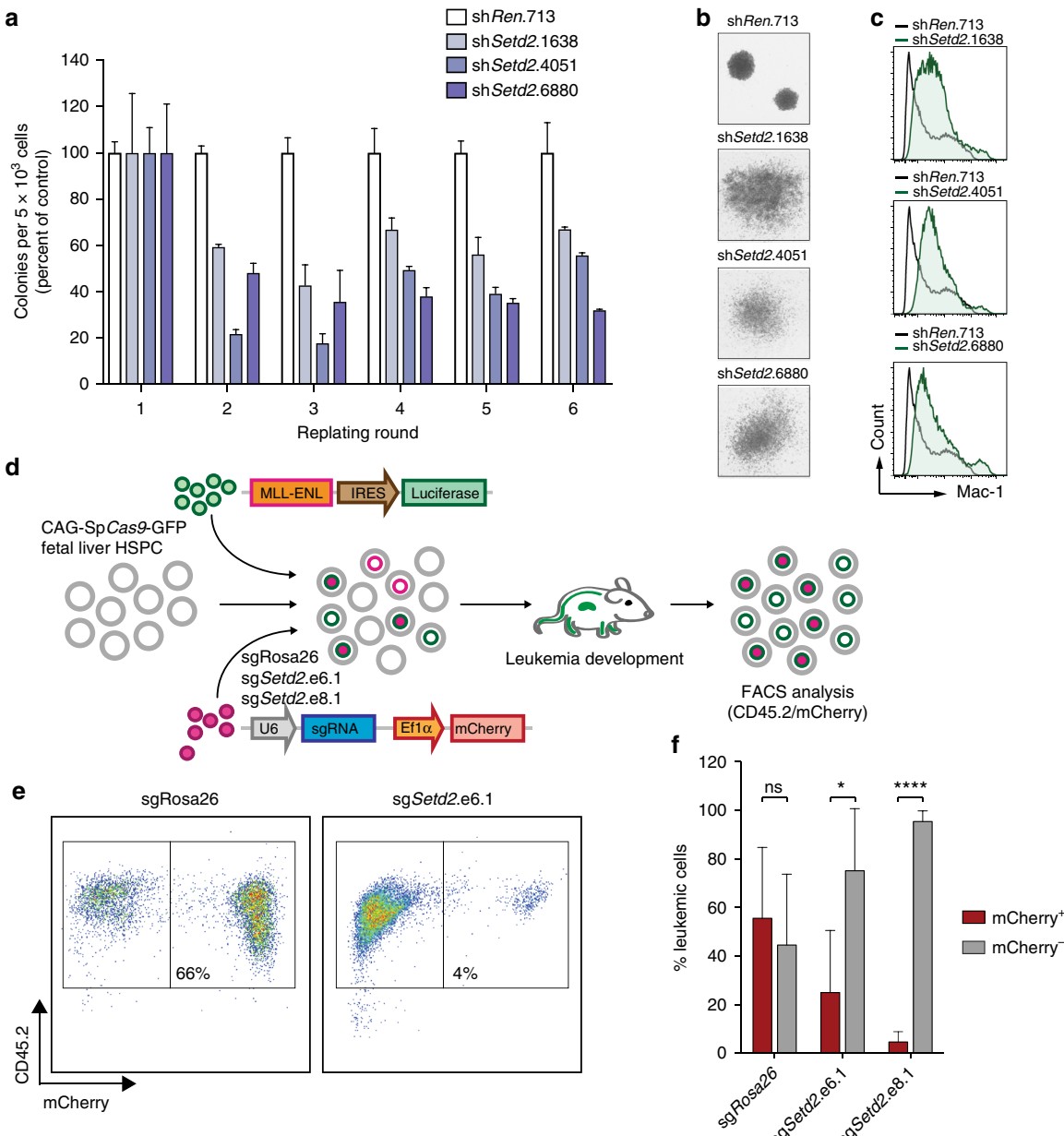

**Fig. 7** SETD2 is required for oncogenic transformation by MLL-fusions. **a** Serial replating assay of primary MLL-AF9-transformed fetal liver cells upon shRNA-mediated knockdown of *Setd2*. Colony numbers were normalized to cells expressing sh*Ren*.713 (mean ± s.d. *n* = 3). **b** Morphology of colonies of MLL-AF9-transformed fetal liver cells upon shRNA-mediated knockdown of *Setd2*. **c** Flow cytometric analysis of Mac-1 expression of MLL-AF9-transformed fetal liver cells upon shRNA-mediated knockdown of *Setd2*. **d** Schematic representation of the in vivo transformation assay. Fetal liver cells from *SpCas9*-transgenic mice were co-transduced with retroviral vectors expressing MLL-ENL and *Setd2*-targeting or control sgRNAs. Cells were transplanted into lethally irradiated C57BL/6 recipient mice. Terminally sick mice were sacrificed and bone marrow cells were analyzed. **e** Representative flow cytometry plots of donor-derived bone marrow cells from mice transplanted with MLL-ENL and a control sgRNA (sg*Rosa*.26, left) or a sgRNA targeting the SET domain of *Setd2* (sg*Setd2* e6.1, right). Live cells were gated. **f** Quantification of flow cytometry analysis of donor-derived bone marrow cells as shown in **e** (mean ± s.d. *n* ≥ 4). ns, not significant, *$p < 0.05$, ****$p < 0.0001$ (*t*-test)

manner[55]. Thus, functional annotation of the core network of MLL-fusion interactors will contribute to establish novel links between MLL-fusion proteins and important cellular processes that had previously not been associated with the biology of MLL-fusion proteins, such as mRNA splicing or protein transport. Given the involvement of these molecular pathways in basic cellular physiology, it is not surprising that more than one third of proteins in the network of conserved MLL-fusion protein interaction partners were identified as essential in recent genome-wide screens[39–42].

To discern MLL-fusion-associated genetic dependencies from essential genes we employed a subtractive shRNA screening approach. Strikingly, the gene encoding the methyltransferase SETD2 was identified as an MLL-fusion-specific hit from this screen with high confidence. shRNA-mediated knockdown as well as CRISPR/Cas9-induced mutagenesis of SETD2 caused proliferation arrest and myeloid differentiation of MLL-fusion-expressing primary and transformed human, and mouse AML cells in vitro and in vivo. This is surprising, because SETD2 has been implied to have tumor suppressor activity in various

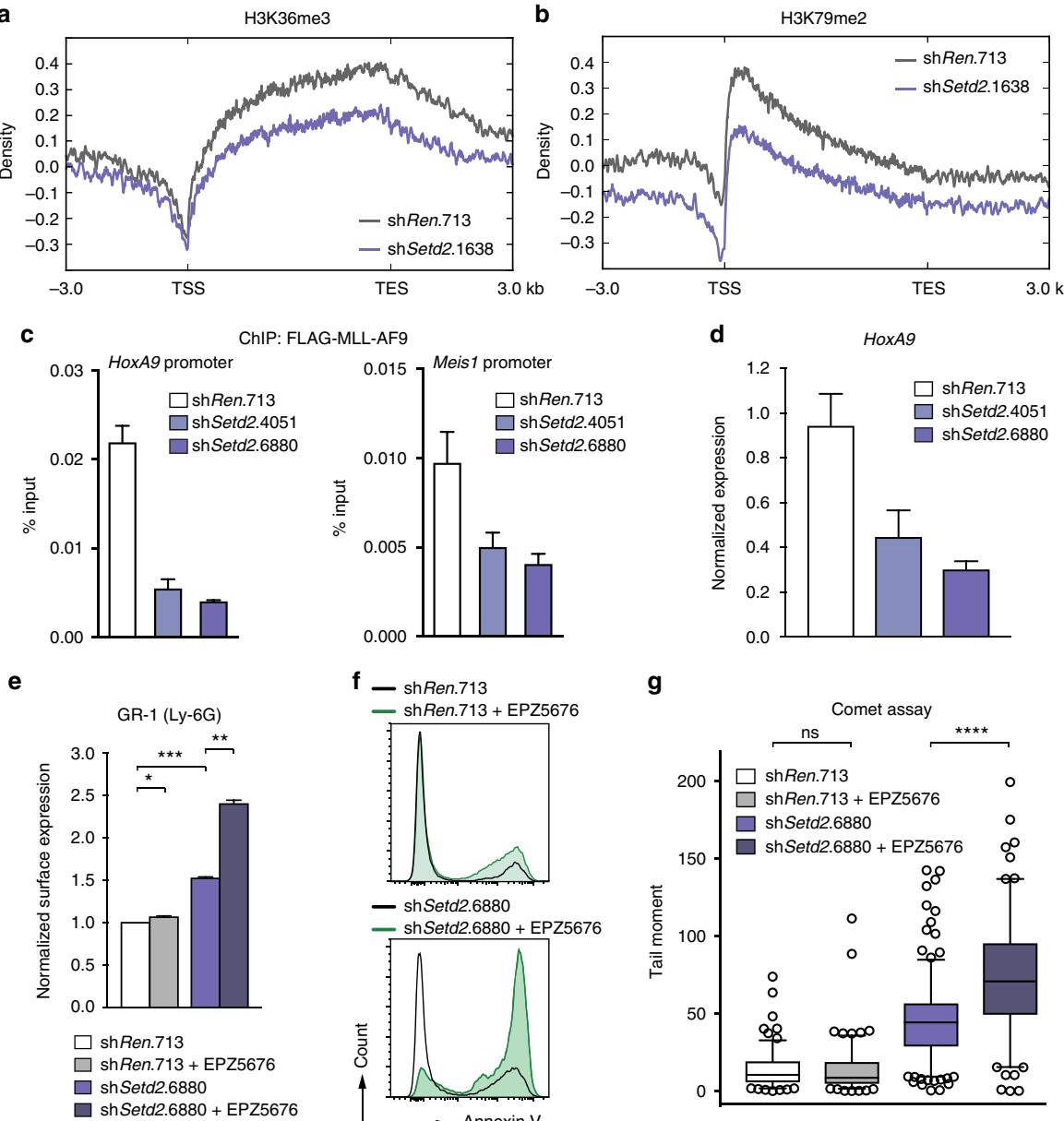

**Fig. 8** SETD2 loss sensitizes AML cells to DOT1L inhibition. Metagene plots of ChIP-Rx data for H3K36me3 (**a**) and H3K79me2 (**b**) after shRNA-mediated knockdown of *Setd2* in mouse *MLL-AF9/Nras*G12D AML cells. **c** qPCR analysis of enrichment on *HoxA9* and *Meis1* promoter regions after MLL-AF9-FLAG-ChIP upon shRNA-mediated knockdown of *Setd2* (mean ± s.d. *n* = 3). **d** qPCR analysis of *HoxA9* mRNA levels in MLL-AF9-FLAG cells expressing indicated shRNAs (mean ± s.d. *n* = 3). **e** Quantification of surface expression of Gr-1 (Ly-6G) after shRNA-mediated knockdown of *Setd2* in mouse *MLL-AF9/Nras*G12D AML cells treated with EPZ5676 (500 nM) (mean ± s.d. *n* = 2). **f** Flow cytometric analysis of Annexin V-positive cells in MV4-11 cells treated with EPZ5676 (50 nM) after shRNA-mediated knockdown of *SETD2*. **g** Quantification of tail moments in an alkaline comet assay performed after shRNA-mediated knockdown of *Setd2* in mouse *MLL-AF9/Nras*G12D AML cells treated with EPZ5676 (500 nM). Quantification of >150 cells is shown. ns, not significant, *$p < 0.05$, **$p < 0.01$, ***$p < 0.001$, ****$p > 0.0001$ (*t*-test)

malignancies, including leukemia[56–58]. SETD2 knockdown was reported to cause a driver-independent proliferative advantage of leukemia cells in vitro and in vivo[58]. In contrast, another report showed that mutational disruption of the *SETD2* SET domain was incompatible with MLL-AF9 AML cell growth[48]. Furthermore, several genome-wide CRISPR/Cas9 screens identified *SETD2* as an essential gene in leukemia cell lines[39–42]. Finally, a recent report showed that while homozygous *Setd2* deletion in the mouse strongly delayed leukemogenesis, heterozygous *Setd2* deletion accelerated MLL-AF9-induced leukemia and caused chemoresistance[47]. This is consistent with increased frequencies of *SETD2* mutations in high-risk leukemia patients that show

increased genomic complexity and chromothripsis[57], and often exhibit therapy resistance and relapse[56]. Therefore, as the majority of cancer patients carry heterozygous *SETD2* mutations, *SETD2* might act as a haplo-insufficient tumor suppressor. In contrast complete loss of SETD2 strongly impedes leukemia development.

The most prominent cellular function of SETD2 is its non-redundant H3K36 tri-methylation activity[44]. The H3K36me3 mark is enriched on gene bodies of actively transcribed genes[59]. We found that MLL-target genes displayed high H3K36me3 levels, validating our proteomic identification of SETD2 as an interactor of MLL-fusion proteins at the genomic level. Given the

large number of proteins harboring a H3K36me3-recognizing PWWP motif[60], several key cellular processes were postulated to be influenced by this epigenetic mark, including transcriptional elongation, splicing, and epigenetic control of gene expression[61]. Consistent with a role for the PWWP motif of the MLL-interactor LEDGF in chromatin binding of MLL-fusion proteins[9], SETD2 loss caused reduced binding of MLL-AF9 to target promoters. Thus, the interaction between SETD2 and MLL-fusion proteins could be required to ensure efficient chromatin binding of MLL-fusions through the maintenance of high H3K36me3 levels on MLL-target genes.

Our results clearly show that SETD2 is involved in the control of the DNA damage response in MLL-fusion-expressing cells. SETD2-deficient cells exhibited high amounts of DNA damage and increased γ-H2AX levels, even in the absence of exogenous genotoxic stress. This is in line with a recent study showing that SETD2 mutations in leukemia impair the DNA damage response, thereby leading to chemotherapy resistance[47]. This defect is attributed to loss of H3K36me3-dependent recruitment of repair proteins to sites of DNA damage. It was shown that MLL-AF9-transformed cells require an intact DNA damage response for full oncogenicity, as experimental induction of DNA damage led to differentiation of leukemia cells[62]. In line with this, down-regulation of SETD2 was sufficient to induce myeloid differentiation of AML blasts. Therefore, a physical and functional interaction between MLL-fusion proteins and SETD2 could be required to guarantee efficient, H3K36me3-dependent repair of DNA lesions that continuously occur during MLL-fusion-induced oncogenic transcription. However, given the accumulation of DNA damage upon SETD2 loss, continuous SETD2 inhibition might result in increased formation of chemoresistant AML subclones.

We found that SETD2 was required for the maintenance of a specific dual H3K36me3-H3K79me2 signature on target genes of MLL-fusions. The H3K79me2 mark is catalyzed by the histone methyltransferase DOT1L, which is critical for the establishment and maintenance of MLL-rearranged leukemia[21]. SETD2 down-regulation rendered MLL-fusion-expressing AML cells hyper-sensitive to the pharmacological DOT1L-inhibitor EPZ5676 (Pinometostat), which is currently in clinical development. It will be interesting to test whether this synergy can be exploited to efficiently target chemoresistant AML cells that are carrying SETD2 mutations.

In summary, our combined proteomic-functional genomic analysis of MLL-fusion protein interactors enabled us to reveal the molecular logic of how modular protein–protein interactions can influence the oncogenicity of MLL-fusion proteins. Our studies provide novel insights into the biology of MLL-fusion proteins and identify an unexpected dependency of MLL-fusion-expressing leukemia cells on the methyltransferase SETD2 during leukemia initiation and maintenance, validating SETD2 as an actionable target MLL-rearranged leukemia.

## Methods

**Constructs**. MLL-fusion genes were assembled by fusing the cDNA of the *MLL* N-terminus (amino acids 1-1396) to C-terminal parts of AF1p (*EPS15*), AF4 (*AFF1*), AF9 (*MLLT3*), CBP (*CREBBP*), EEN (*SH3GL1*), ENL (*MLLT1*), and GAS7 (*GAS7*) and cloned into pcDNA5/FRT/TO/SH/GW. Generation of the miR-E shRNA vectors RT3GEN and RT3REN was previously described[63,64]. The SEM cell line was infected with SGEN[64]. A pMSCV-MLL-AF9-IRES-Venus construct was used for the in vitro re-plating assay, while a pMSCV-MLL-ENL-IRES-Luc2 construct was used for the in vivo transformation assay. A V5-tagged version of the N-terminal part of MLL (amino acids 1-1396) was cloned into a vector containing a Doxycycline-inducible promoter. The C-terminal fragment of SETD2 (amino acids 950-2570) was cloned with a N-terminal 6×-Myc tag. The library of 768 shRNAs was designed to target 128 conserved interaction partners of ≥5 selected MLL-fusions with six shRNAs per gene. 97-mer oligomers (Integrated DNA Technologies) were reconstituted in $H_2O$ and stored at −80 °C. Mini-pools of six shRNAs

targeting the same candidate gene were amplified in parallel PCR reactions using Pfx DNA polymerase (Invitrogen) as described[64]. Reactions were pooled and purified using PCR Clean-up kit (Qiagen). PCR products were digested with EcoRI and XhoI (New England Biolabs) and ligated with retro- or lentiviral vectors allowing for inducible or constitutive shRNA expression together with selection markers. After dialysis, ligations were introduced into Mega X DH10ß T1 electro-competent cells (Invitrogen) by electroporation (2 kV, 200 Ω, 25 μF) using a MicroPulser Electroporator (Bio-Rad). The library was purified using Midi Prep Kit (Qiagen). The presence of shRNA cassettes was verified by Sanger sequencing. For CRISPR/Cas9-mediated mutagenesis, sgRNAs were cloned into lentiviral vectors allowing for constitutive sgRNA expression together with GFP or mCherry as previously described[65]. Sequences of sgRNAs used in the study are listed in Supplementary Table 2.

**Cell culture**. All standard human leukemia cell lines such as: MOLM13, MV4-11, HEL, HL-60, KYO-1, were obtained from DSMZ (Deutsche Sammlung von Mik-roorganismen und Zellkulturen GmbH (DSMZ, www.dsmz.de)) or the American Type Culture Collection (ATCC, www.atcc.org) and modified to express the eco-tropic receptor and rtTA3. The murine Tet-On *MLL-AF9/Nras*G12D AML cell line (RN2) was previously described[66]. All cell lines were cultured in RPMI 1640 (Gibco) supplemented with 10%FBS, 100 U/ml penicillin, and 100 μg/ml strepto-mycin. Platinum-E cells were maintained in DMEM (Gibco) supplemented with 10% FBS, 100 U/ml penicillin, and 100 μg/ml streptomycin. *SpCas9*-expressing variants of MOLM-13 and *MLL-AF9/Nras*G12D cells were generated by lentiviral transduction followed by selection with Blasticidin (10 μg/ml). The *SpCas9*-expressing subclone of MV4-11 was a gift from G. Winter (Dana Farber Cancer Institute, Harvard University). The *SpCas9*-expressing THP-1 and NOMO-1 cell lines were previously described[67]. MLL-AF9-FLAG cells were previously descri-bed[22]. For proliferation curves, cells were seeded at low densities in triplicates and cell numbers were determined using a multi-channel electronic cell counter (CASY-I; Omni Life Science) in regular intervals. The DOT1L inhibitor EPZ5676 was obtained from BPS Bioscience. Human leukemic blast cells from heparinized samples of AML patients (*n* = 3) were isolated on Ficoll-Hypaque gradients and stored in liquid nitrogen. After thawing, cells were cultured in RPMI 1640 medium containing 10% BIT 9500 Serum Substitute, 100 ng/ml SCF, 50 ng/ml Flt3L, 20 ng/ml IL-3, 20 ng/ml G-CSF (all PeproTech), $10^{-4}$ M ß-mercaptoethanol, 50 μg/ml gentamicin, and 10 μg/ml ciprofloxacin plus 500 nM SR1 and 1 μm UM729[68]. This protocol typically leads to sustained proliferation of primary human AML cells over 20 days, yielding a >10-fold expansion in vitro. All patients gave written informed consent before blood or bone marrow was obtained. The study was approved by the Institutional Review Board of the Medical University of Vienna. Personal data from AML patients were used according to ethics approvals of clinical partners for collection of clinical and genetic data upon informed consent. All cell lines have been tested for mycoplasma contamination. Cell lines used in this study were not listed in the database of commonly misidentified cell lines main-tained by ICLAC.

**Viral transduction**. For retroviral transductions, Platinum-E cells were transiently transfected with pGAG-POL and retroviral expression vectors using the calcium-phosphate method in the presence of Chloroquine (25 μm, Sigma-Aldrich). Virus-containing supernatant was harvested, filtered (0.45 μm), and supplemented with polybrene (5 μg/ml). Target cells were spinoculated at 1300×*g* for 90 min. For lentiviral transductions, HEK293T cells were transiently transfected with psPAX2, pMD2.G, and lentiviral expression vectors. Virus-containing supernatant was harvested, filtered (0.45 μm), and supplemented with polybrene (5 μg/ml). Target cells were spinoculated at 1300×*g* for 90 min. Human primary AML cells were transduced with concentrated lentiviral supernatants via centrifugation (1200×*g*, 90 min) at a multiplicity of infection of 20.

**Generation of Flp-In cell lines**. Jurkat Flp-In cells (Invitrogen) were transduced with pLenti6/TR (Thermo) and a clone expressing high levels of the tetracycline repressor (TR) was isolated. Cells were transfected with targeting constructs (in pcDNA5/FRT/TO) together with pCAAGS-Flp-E by nucleofection using program X-001 (Amaxa). Targeted cells were selected in Clonacell TCS medium (Stem Cell Technologies) supplemented with 600 μg/ml Hygromycin B. Clones were isolated and expanded in liquid medium in the presence of Hygromycin B. Expression of MLL-fusions was tested by qRT-PCR after induction of transgene expression by addition of 1 μg/ml Doxycycline for 24 h.

**Affinity purification of protein complexes**. Nuclear extracts from transgene-expressing Jurkat cells were prepared and single-step STREP-Tactin purifications of MLL-fusion proteins were performed as described[33]. All purifications of MLL-fusion proteins were performed from $1 \times 10^9$ freshly harvested cells. After being washed with PBS, cells were incubated in buffer N (300 mM sucrose, 10 mM HEPES pH 7.9, 10 mM KCl, 0.1 mM EDTA, 0.1 mM EGTA, 0.1 mM DTT, 0.75 mM spermidine, 0.15 mM spermine, 0.1% Nonidet P-40, 50 mM NaF, 1 mM $Na_3VO_4$, protease inhibitors) for 5 min on ice. Nuclei were collected by cen-trifugation (500×*g* for 5 min), and the supernatant was removed. The nuclear pellet was washed with buffer N. For the extraction of nuclear proteins, nuclei were

resuspended in buffer C420 (20 mM HEPES pH 7.9, 420 mM NaCl, 25% glycerol, 1 mM EDTA, 1 mM EGTA, 0.1 mM DTT, 50 mM NaF, 1 mM $Na_3VO_4$, protease inhibitors), vortexed briefly, and shaken vigorously for 30 min. After centrifugation for 1 h at 100,000×g, the protein concentration of the soluble nuclear fraction was measured by Bradford assay. Prior to purification, all nuclear extracts were adjusted to 2 mg/ml and 150 mM NaCl with HEPES buffer (20 mM HEPES, 50 mM NaF, 1 mM $Na_3VO_4$, protease inhibitors). 15 mg of nuclear extract were pre-treated with benzonase (20 U/ml) and RNase A (50 ng/ml) for 15 min at 4 °C. Nonspecific binding to the affinity resin was blocked by the addition of avidin (1 μg/ml). 150 μl StrepTactin sepharose (IBA) was added and lysates were incubated for 2 h at 4 °C with agitation. Beads were washed 3 times with TNN-HS buffer (50 mM HEPES pH 8.0, 150 mM NaCl, 5 mM EDTA, 0.5% NP-40, 50 mM NaF, 1 mM $Na_3VO_4$, and protease inhibitors). Bound proteins were eluted by the addition of 100 μl 2.5 mM Biotin (Alfa Aesar) in TNN-HS buffer. Samples were digested with trypsin and processed for LC-MS/MS analysis.

**Mass spectrometry.** Analysis of affinity purification samples was performed as described previously[33,34]. All affinity purifications were analyzed on a hybrid linear trap quadrupole (LTQ) Orbitrap Velos mass spectrometer (Thermo Fisher Scientific) coupled to a 1200 series high-performance liquid chromatography system (Agilent Technologies) via a nano-electrospray ion source using liquid junction (Proxeon). Solvents for HPLC separation of peptides were as follows: solvent A consisted of 0.4% formic acid in water, and solvent B consisted of 0.4% formic acid in 70% methanol and 20% isopropanol. 8 μl of the tryptic peptide mixture were automatically loaded onto a trap column (Zorbax 300SB-C18 5 μm, 5 × 0.3 mm, Agilent Biotechnologies). After washing, peptides were eluted by back-flushing onto a 16-cm-fused silica analytical column with an inner diameter of 50 μm packed with C18-reversed phase material (ReproSil-Pur 120 C18-AQ, 3 μm, Dr. Maisch) with a 27-min gradient ranging from 3 to 30% solvent B, followed by a 25-min gradient from 30 to 70% solvent B and, finally, a 7-min gradient from 70 to 100% solvent B at a constant flow rate of 100 nl/min. Analyses were performed in a data-dependent acquisition mode, and dynamic exclusion for selected ions was 60 s. A top 15 collision-induced dissociation (CID) method was used, and a single lock mass at $m/z$ 445.120024 ($Si(CH_3)_2O)_6$) was employed. Maximal ion accumulation time allowed in CID mode was 50 ms for $MS^n$ in the LTQ and 500 ms in the C-trap. Automatic gain control was used to prevent overfilling of the ion traps and was set to 5000 in $MS^n$ mode for the LTQ and $10^6$ ions for a full FTMS scan. Intact peptides were detected in the Orbitrap Velos at 60,000 resolution at $m/z$ 400.

**Protein identification and network analysis.** For protein identification, raw MS data files were converted into Mascot generic format (.mgf) files and searched against the human SwissProt protein database (v. 2013.01) using the two search engines Mascot (v2.3.02, MatrixScience, London, UK) and Phenyx (v2.6, GeneBio, Geneva, Switzerland). Carbamidomethyl cysteine and oxidized methionine were set as fixed and variable modifications, respectively; one missed tryptic cleavage site per peptide was permitted. The Mascot and Phenyx identifications were combined and filtered as described[32] to provide <1% protein false discovery rate (FDR). Known MS contaminants, such as trypsin and keratin were removed from the results, and further analysis of proteins specifically binding to the baits was achieved by fitting the MS data to the generalized linear statistical model: log(data) $\sim A_{0,j} + A_{i,j} + \alpha_i + \beta_{i,k}$, where $A_{0,j}$ is the logarithm of the baseline abundance of the $j$-th prey protein (estimated from the control AP-MS experiments), $A_{i,j}$ is the specific enrichment of the $j$-th prey in the pulldowns of $i$-th bait, and $\alpha_i$ and $\beta_{i,k}$ are the normalization terms that model the abundance of background proteome in the $k$-th replicate pulldown of $i$-th bait (to estimate $\alpha_i$, DDX5 and DDX17 proteins were used, as these are known components of the nuclear proteome background and were ubiquitously present in all AP-MS experiments). To improve the accuracy, the model was independently applied to three different types of MS data: Protein spectral counts (the Poisson distribution was used to model the data) and the sum of peptide scores from either Mascot or Phenyx search results, assuming the log-normal distribution. Only peptides unique to the protein groups were used. The inference of the model parameters was achieved using JAGS v.3.0. For each type of MS data, the $p$-value for the hypothesis that $A_{ij} > 0$ (i.e., that the $j$-th prey binds specifically to the $i$-th bait) was calculated and then the three $p$-values were combined into a single $p$-value using the Fisher method. All the identified bait-prey pairs were ranked by the combined $p$-value. The 300 most significant interactions per bait were retained. This cutoff represents a compromise between ensuring high statistical significance of the included interactors, but also capturing sufficient diversity in the interactomes of the selected MLL-fusion proteins. The proteins shared by at least five baits were selected for further analysis. Seven proteins were manually removed as these were either frequently observed contaminants or were not detected in human hematopoietic cell lines. The final set was comprised of 128 proteins. The resulting network was extended by the known protein–protein interactions, which were retrieved from three different datasets: (i) the set of non-redundant complexes in CORUM[69], from which binary protein–protein interactions were extracted using the matrix model. (ii) The set of interactions described in ref. [70], which combines data from several public repositories. (iii) The set of interactions reported in ref. [71], integrating different data sets. After removing self-interactions, the final network consisted of 365 PPIs between 101 core MLL-fusion interactors, while 27 other interactors identified by AP-MS remained connected

only to the MLL-fusion baits. The network was partitioned into distinct protein communities by maximizing the modularity score of the network over all possible partitions using the "cluster_optimal" function of the "igraph" package in R. Gene ontology (GO) term enrichment analysis for each separate network community was performed. Enrichment was computed with the topGO package from R, using the default algorithm and the annotation file from geneontology.org (18 November 2015). All human proteins in UniProtKB/Swiss-Prot were used as the background population. $p$-values were corrected for multiple testing using the Benjamini–Hochberg procedure (FDR). Based on functional annotation similarity, unconnected nodes were assigned to the most enriched GO terms in each community. Enrichment of protein complexes within the network of 960 MLL-fusion-interactors was estimated by Fisher's exact test. Before enrichment, CORUM core complexes sharing >70% of the proteins were iteratively merged to reduce the redundancy. All proteins present in at least one complex were used as the background population. $p$-values were corrected for multiple testing as explained above. Protein interaction networks were visualized using Cytoscape and Gephi. Detailed information about all 960 identified interactors of MLL-fusion proteins is provided in Supplementary Data 1.

**Negative selection RNAi screening.** MOLM-13 cells transduced with mini-pools of retroviral vectors for shRNA-mediated targeting of conserved MLL-interactors (coupled to GFP) were mixed in a 50:50 ratio with cells expressing control shRNAs (coupled to dsRed) and cultured in the presence of Doxycycline (1 μg/ml). Changes in GFP/dsRed ratios were examined by flow cytometry over time. Percentages of GFP-positive cells were measured at each time point during the experiment and normalized to initial measurement after 2 days of Dox treatment. Gene essentiality was assessed based on recent large-scale datasets from genome-wide screens[39–42]. Based on individual scores from single screens, we assigned scores of 1 (essential) vs. 0 (non-essential) to each gene in our data set. Our combined essentiality score reflects the sum of all essentiality information per gene from 18 different experiments. Thus, a gene that is ubiquitously essential will obtain a score of 18, while a ubiquitously non-essential gene will obtain a score of 0. A gene was called essential if it scored in ≥10 of 18 cell lines. Sequences of shRNAs used for the RNAi screen are listed in Supplementary Data 2 and Supplementary Table 3.

**Chromatin immunoprecipitation (ChIP-Rx) and sequencing.** MLL-AF9/NrasG12D AML cells and *Drosophila melanogaster* S2 cells were separately cross-linked with 10% formaldehyde and quenched with glycine (2.5 M). Pellets were washed, pooled, and resuspended in SDS lysis buffer (1% SDS, 10 mM EDTA, 50 mM Tris-HCl, pH 8.0). Chromatin was sonicated to obtain fragments of 150 bp using a Bioruptor sonicator (Diagenode). 0.5% Triton X-100 was added to the samples to allow solubilization of the sheared DNA. Chromatin was incubated with antibodies overnight (5 μg each). Antibody-bound material was recovered using protein-G-coupled magnetic beads (Invitrogen), washed (50 mM Hepes-KOH, pH 7.4; 500 mM LiCl; 1 mM EDTA; 1% NP40 and 0,5% Na-Deoxycholate), and released using elution buffer (50 mM Tris-HCl, pH 8.0; 10 mM EDTA and 1% SDS) at 65 °C. DNA-protein crosslinks were reverted by incubating the samples overnight at 65 °C in the presence of 0.2 M NaCl. The DNA was treated with RNaseA (0.2 mg/ml) and proteinase K (0.2 mg/ml) and purified using PCR clean-up kit (Qiagen). Chromatin immunoprecipitation of FLAG-tagged MLL-AF9 was performed using the High Sensitivity ChIP Kit (Abcam, 185913) according to the manufacturer's instructions. Antibodies used were: anti-H3K4me3 (Abcam, 8580) anti-H3K36me3 (Abcam, 9050), anti-H3K79me2 (Abcam, 3594), anti-Flag (Sigma, F1804). Sequencing libraries were prepared using NEBNext Ultra DNA Library Prep Kit for Illumina (New England BioLabs) and sequenced on Illumina HiSeq 4000 using 50 bp single-read chemistry.

Raw ChIP-seq reads were evaluated with FastQC (version 0.11.4). Quality-filtering and trimming was done with PRINSEQ-lite (version 0.20.4). Resulting high-quality reads were simultaneously mapped against the *Mus musculus* (GRCm38) and *Drosophila melanogaster* (dm6) reference genomes via BWA (version 0.7.15). SAMtools (version 1.4) was used to split the alignments into mouse and Drosophila reads. Read normalization via the *Drosophila melanogaster* spike-in material was carried out with Deeptools (version 2.5.0.1) for each sample. Profile plots of histone marks were also generated with Deeptools (version 2.5.0.1). For the comparison of H3K79me2 vs. H3K36me3 signal intensities on MLL-target genes vs. non-MLL-targets, an equally sized set of randomly selected non-MLL-target genes was chosen. MLL-target genes represent genes that were downregulated upon MLL-AF9 withdrawal as measured by microarray analysis[22]. IGV was used for manual inspection and visualization of data. For the analysis of histone mark intensities in genes, mapped reads per gene were counted with featureCounts (1.5.0), respective input counts subtracted, and normalized via TMM using the edgeR package. The Pearson correlation coefficient between changes in respective histone marks over gene bodies after *Setd2* knockdown was calculated with the functions bigwigCompare, multiBigwigSummary, and plotCorrelation of Deeptools.

**RNA sequencing.** RNA was isolated using RNeasy kit (Qiagen). The amount of total RNA was quantified using the Qubit 2.0 Fluorometric Quantitation system (Life Technologies) and the RNA integrity number was determined using the

Experion Automated Electrophoresis System (Bio-Rad). RNA-seq libraries were prepared with TruSeq Stranded mRNA LT sample preparation kit (Illumina) using Sciclone and Zephyr liquid handling robotics (PerkinElmer). Sequencing libraries were pooled, diluted, and sequenced on an Illumina HiSeq 3000 using 50 bp single-read chemistry. Base calls provided by the Illumina Realtime Analysis software were converted into BAM format using Illumina2bam and demultiplexed using BamIndexDecoder (https://github.com/wtsi-npg/illumina2bam). Initial quality control of raw sequencing reads was done with FastQC (version 0.11.4) followed by pre-processing with PRINSEQ-lite (version 0.20.4). Resulting high-quality reads were mapped via STAR[72] (version 2.5.0b) against the mouse (GRCm38) reference genome. After processing of the alignment results with SAMtools (0.1.19) counts per gene were obtained by HTSeq[73] (version 0.6.0). Normalization and differential expression analysis between two samples was carried out with DESeq2[74]. For the visualization of gene expression and unsupervised hierarchical clustering of the samples the rlog normalization in DESeq2 was applied. We used the R library pheatmap for sample clustering (euclidian distance, complete linkage clustering) and heatmap.2 from the gplots package to visualize differentially expressed genes (Pearson correlation and ward.D clustering).

**Real-time PCR analysis.** Total RNA was isolated using RNeasy mini kit (Qiagen). Reverse transcription was performed with RevertAid RT Kit (Thermo Scientific) using 300 ng RNA. Quantitative PCR was performed using SensiMix SYBR Hi-ROX kit (Bioline) on a RotorGene Q PCR machine (RG-600, Qiagen). Results were analyzed using the $2^{-ddC(t)}$ method. Sequences of primers used for qPCR are listed in Supplementary Table 4.

**FACS analysis.** Cells were incubated in Fc block reagent (murine: Biolegend,14-0161-85, clone93; human: Biolegend 422301) prior to incubation with the following antibodies: anti-human CD36 (Biolegend, 336207, clone 5-271), Brilliant Violet 421 anti-mouse/human CD11b (Biolegend, 101235, clone M1/70), APC Rat anti-Mouse CD117 (BD Pharmingen, 553356 = cell, clone 2B8), anti-Mouse Ly-6G (Gr-1) (Biolegend, 108411, clone RB6-8C5). Samples were measured on LSR Fortessa or Canto II flow cytometers (BD Biosciences) and analyzed using FlowJo software (Tree Star).

**Co-immunoprecipitation.** HEK293 cells stably expressing pMSCV-rtTA3-IRES-EcoR-PGK-Puro were transiently transfected with the indicated constructs using the PEI transfection method. The expression of the N-terminal MLL-fragment was induced with Doxycycline for 24 h (1 µg/ml, Sigma Aldrich). The Proteasome inhibitor MG-132 (Sigma-Aldrich, 10 mM in DMSO), was added to the medium 2 h before cell harvest. Cells were harvested in IP-lysis buffer (50 mM Tris/HCl pH 7.5, 150 mM NaCl, 1% NP-40, 5 mM EDTA, 5 mM EGTA) supplemented with protease and phosphatase inhibitors. Protein concentrations were determined with Bradford protein assay (Biorad) using γ-globulin (Biorad) as a standard. Subsequently, lysates were incubated with Anti-Myc-tag mAb-Magnetic Beads (Biomedica GmbH) for 1.5 h with continuous rotation at 4 °C. Beads were recovered by centrifugation and washed in IP-buffer. Bound proteins were released by addition of Lämmli-sample-buffer (Biorad) and boiling for 10 min at 95 °C before SDS-PAGE analysis and immunoblotting.

**Western blotting.** Western blotting was performed according to standard laboratory protocols. Antibodies used were: anti-H3K36me3 (Abcam, 9050; 1:1000), anti-H3 (Abcam, 1791; 1:5000), anti-HA (Covance, MMS-101P; 1:2000), anti-H2AX (Millipore 05-636; 1:5000) anti-Tubulin (Abcam, 7291; 1:5000), anti-RCC-1 (Santa Cruz, sc-55559; 1:2000), anti-p21 (Santa Cruz, sc-6246; 1:1000), anti-GAPDH (Santa Cruz, sc-365062; 1:1000), anti-V5-tag (Cell Signaling, 13202; 1:2000), anti-Myc (Abcam, 9106; 1:10000). Secondary antibodies used were: goat anti-mouse HRP (Jackson ImmunoResearch, 115-035-03 or Thermo Fisher Scientific, 31430; 1:5000), goat anti-rabbit HRP (Jackson ImmunoResearch, 111-035-003 or Thermo Fisher Scientific 31460; 1:5000). Uncropped scans of all blots are shown in Supplementary Fig. 12.

**Cytospin analysis.** Cells were cytocentrifuged onto glass slides and stained with Giemsa staining solution before microscopic analysis. Images were processed using Adobe Photoshop (Adobe).

**Comet assay.** Cells were treated with Doxycycline (1 µg/ml) to induce shRNA expression. shRen.713-expressing cells were treated with 150 µm $H_2O_2$ for 10 min. $4 \times 10^4$ cells were washed in PBS and mixed with 100 µl, 0.5% low melting agarose. The cell suspension was deposited on pre-chilled frosted glass slides pre-coated with 1% agarose. Slides were immersed in pre-chilled lysis buffer (2.5 M NaCl, 10 mM Tris-HCl, 100 mM $Na_2$EDTA, 10% DMSO, and 1% Triton X; pH 10) for 1–2 h and washed with cold $H_2O$ (3 times for 10 min). Slides were incubated in electrophoresis buffer (55 mM NaOH, 1 mM EDTA, 1% DMSO; pH 12.8) for 45 min followed by electrophoresis at 35 V for 40 min. Samples were neutralized in 400 mM Tris-HCl buffer pH 7.0 for 1 h and washed once with pre-chilled $H_2O$ before staining with SYBR Gold. Comet tail moments, defined as the average distance

migrated by the DNA multiplied by the fraction of DNA in the comet tail, were scored using the CASP image-analysis software.

**Transplantation experiments.** $1 \times 10^6$ Murine *MLL-AF9/NrasG12D* cells were injected into the tail-vein of sub-lethally (5.5 Gy) irradiated C57BL/6 Ly5.1 recipient ($n = 5$). Disease progression was monitored by bioluminescence imaging. Doxycycline (4 mg/ml) was supplied to the drinking water of mice to activate the expression of shRNAs. E14.5 fetal liver cells from C57BL/6 Ly5.2 embryos with heterozygous expression of the *SpCas9* transgene[49] were co-transduced with retroviral vectors allowing for constitutive expression of MLL-ENL and Luciferase, and sgRNAs coupled to mCherry. The efficiency of infection ranged between 5–11%. Transduced fetal liver cells were injected into the tail-vein of lethally ($2 \times 5.5$ Gy) irradiated C57BL/6 Ly5.1 recipient mice. Terminally sick animals were sacrificed after 50–60 days, and bone marrow was isolated from femurs and tibia. Animals suffering from obvious other symptoms than leukemia were excluded from the analyses. During all animal experiments we adhered to the 3 R principles (reduction, replacement, and refinement). Animal numbers were determined by the investigator using previous experience and based on judgement of pilot experiments. In general, animal numbers were chosen to be as small as possible but large enough to provide needed estimates for statistical tests, based on previous experience. All animal experiments were performed according to ethical animal license protocols approved by the authorities of the Austrian government. No randomization was used in transplantation experiments. The investigator was not blinded to the group allocation during the experiment and while assessing the outcome.

**Hematopoietic progenitor re-plating assay.** Fetal liver cells were retrovirally co-transduced with MLL-AF9 coupled to Venus and vectors allowing for constitutive expression of shRNAs coupled to mCherry. Venus/mCherry double-positive cells were isolated by FACS sorting and seeded in complete methylcellulose medium (MethoCult M3434). Colonies were scored in 7-day intervals and $5 \times 10^3$ cells were re-plated.

**Apoptosis assays.** Annexin V staining was performed according to the manufacturer's protocol (Annexin V Apoptosis Detection Kit PE, Affymetrix, eBioscience). The TUNEL assay was performed according to the manufacturer's instructions (ApoBrdU Red DNA Fragmentation Kit; BioVision (K404-60)). Cells were analyzed by flow cytometry.

**Genotyping of cells with CRISPR/Cas9-induced mutations.** Targeted regions were amplified in a PCR reaction using LA Taq® DNA Polymerase (TaKaRa RR002A). PCR products were purified (Qiagen) and analyzed by Sanger sequencing. Chromatograms were analyzed with the TIDE tool (Tracking of Indels by Decomposition, https://tide-calculator.nki.nl)[75] to quantify nature and frequency of generated indels.

**Cell cycle analysis.** Murine *MLL-AF9/NRas*G12D AML cells were cultured in the presence of Doxycycline (1 µg/ml), harvested, fixed in 70% Ethanol and stored at −20 °C until further analysis. Cells were stained with PI staining solution and examined by flow cytometry.

**Statistical analysis.** Two-tailed Student's $t$-tests were used for statistical analysis if not stated otherwise.

**Data availability.** The mass spectrometry proteomics data have been deposited to the ProteomeXchange Consortium via the PRIDE partner repository with the dataset identifier PXD009338. RNA-seq and ChIP-seq data was deposited into the Gene Expression Omnibus (GEO). GEO accession GSE110521.

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

## Acknowledgements

We thank G. Winter for Sp*Cas9*-expressing MV4-11 cells and J. Loizou, V. Sexl, and A. Villunger for critical advice on the manuscript. We also thank all members of the Grebien and Superti-Furga laboratories for discussions. We are grateful for the expert technical assistance of A. Mazouzi with comet assays. We thank G. Stefanzl for storing and preparing primary leukemia cells, A. Spittler and G. Hofbauer (Core Facility Flow Cytometry, Medical University of Vienna) for cell sorting, the IMP animal facility staff for animal caretaking and the Biomedical Sequencing Facility at CeMM for assistance with next-generation sequencing. N. Krall is acknowledged for preparing ethics application 1676/2016 for work with biobanked samples. A.S. was supported by the European FP7 Marie-Curie Initial Training Network HEM_ID. B.G. was supported by the NVKP_16-1-2016-0037 grant of the National Research, Development and Innovation Office, Hungary. C.R.V. is supported by the National Institutes of Health grant NCI RO1 CA174793, NCI 5P01CA013106-Project 4, and a Leukemia & Lymphoma Society Scholar Award. Research in the Valent laboratory is supported by the Austrian Science Fund (FWF) F4704. Research in the Zuber laboratory is supported by a SFB grant (F4710) of the Austrian Science Fund (FWF), and generous institutional funding by Boehringer Ingelheim. Research in the Superti-Furga Laboratory is supported by ERA-NET Grant I 2192-B22, the FWF Stand-Alone Grant P 29250-B30, and FWF SFB Grant F 4711-B20. Work in the Grebien laboratory is supported by grant no. 857935 from the Austrian Research Promotion Agency (FFG) to J.E. This project has received funding from the European Research Council under the European Union's Horizon 2020 research and innovation programme (grant agreement n° 336860/StG to J.Z., grant agreements n° 695214/AdG, and n° 727416/PoC to G.S.F. and grant agreement n° 636855/StG to F.G.).

## Author contributions

A.S., J.E., J.S., M.R., S.V., and F.G. planned and performed experiments and analyzed results. T.E., A.C.R., A.St., and B.G. performed bioinformatic analyses. M.R., M.M., J.J., M.A., B.L., C.V., and J.Z. provided protocols, cell lines, and vectors used in the study. A.C.M and K.L.B performed MS analysis, planned, and performed experiments and analyzed results. P.V. provided access to patient materials. A.S., J.Z., G.S.F., and F.G. designed the study, planned experiments, analyzed results, and wrote the paper.

## Additional information

**Competing interests:** The authors declare no competing interests.

