## [Peer Review File · Nature Communications]

Reviewers' comments:

Reviewer #3 (Remarks to the Author):

Review Skucha et al:

This manuscript has been evolving since its last submission and it has improved. Yet, the whole gist of the manuscript still is not adequately adjusted to the novelty that is conveyed here.

The major points not properly acknowledged in this manuscript are as follows:

- It has been already shown that MLL fusions need the H3K36me binding (PWWP) domain of LEDGF for function (see PMID:18598942, and PMID:24465000 where the abstract reads: "Here we show that only two functional modules are necessary and sufficient for target recognition: those that bind to non-methylated CpGs and di-/tri-methylated histone H3 lysine 36 (H3K36me2/3).")
- A complete loss of Setd2 is NOT incompatible with leukemia development as the authors erroneously indicate. See Mar et al. (PMID:29018079) figure 3C. A clean, genetic deletion of Setd2 delays leukemogenesis but it does not block it completely.

Therefore I'd kindly advise the authors to modify their manuscript as follows (mostly text/display issues)

- Introduction, line 56: Please, correct your statement and acknowledge that it has been shown that menin is necessary to form a compound binding site (together with MLL) for LEDGF and that a direct fusion of the LEDGF-H3K36 reader domain (PWWP) to MLL can replace menin altogether.
- Introduction, line 78 to 82: " Thus, despite...". Delete this sentence as it is incorrect. Of course "shared mechanisms of transformations" have been demonstrated! What about the menin inhibitors?
- Results line 106: It needs to be acknowledged in the text and directly in the header of figure 1d that Jurkat T-cells were used here (write Jurkat, not "human leukemia line"). Do not detract from the fact that T-cells are a somewhat odd choice because T-ALL is almost never affected by 11q23 translocations. Also in line 108 it needs to be figure 1e not "1d".
- Results after line 130: A disclaimer is necessary here stating CLEARLY that most of the interactions identified will be indirect interactions. It is simply impossible from sterical considerations that MLL-N may interact directly with 128 different proteins.
- Results line 173, suppl. figure 3c: Check your labeling. As displayed the logic of the control is wrong! The idea of the control is to show that a myc-specific precipitation will not unspecifically bring down V5-MLL in the absence of myc-Setd2. As shown the control doesn't make sense. Clearly, if V5-MLL is not there from the beginning it won't be precipitated. Either the experiment was set up the wrong way or the blots have been switched.
- Results line 197: There's no "strong" anti-proliferative response. After three weeks the Setd2 k.d. cells still proliferate. A 1.5log difference in cell numbers after three weeks is significant but can be hardly called strong.

- Results line 198, fig 3d and similar: It is mandatory to show real data instead of "relative" heatmaps here. A simple line graph would be ok. It makes a difference if the GFP-positive ratio drops from 50% to 5% or from 50% to 40%. The heatmap only gives an undefined "max" and "min". I don't understand the refusal of the authors to show the actual data.
- Results line 203, fig. 3e: Nothing can be really derived from a single NPMmut sample. It may be just a coincidence. In the absence of a solid and statistically valid group this is a mere chance observation. Please, either get more samples or take out this sub-figure.
- Results line 220, fig.4c,d: Common consensus is that H3K79me2 and H3K36me3 are always and 100% correlated. The authors should show an additional example of a gene where this is not the case. Judging from the signal/noise ratio of the ChIP-seq results shown in fig. 4d, is it possible that for some genes the H3K36me/K79me signals disappear in the background and therefore the correlation is not complete?
- Results line 265: This would be a good place to insert a disclaimer that Setd2 knockout cells still can be transformed by MLL-AF9 although with a prolonged latency.
- Results line 289: Change heading because Setd2 is not absolutely required for oncogenic transformation by MLL-fusion proteins. (see Mar et al).
- Discussion: Please, check your discussion and take out any statement that would indicate that Setd2 is absolutely necessary for transformation by MLL fusions. Setd2 "accelerates" leukemogenesis but it is not essential as shown by genetically clean systems.

In summary the authors do bring up a novel and interesting idea worth publishing. The fact that MLL should contact the very enzyme that deposits the modification that is read by MLL (through LEDGF) is new and very plausible. This situation is with precedent as the known MLL complex (COMPASS) member WDR5 also reads H3K4me that is deposited by wt-MLL. However, the fact that perturbation of H3K36me slows down transformation was highly anticipated as the importance of H3K36me for targeting MLL fusions has been prominently published before.

Editorial note: We asked Reviewer#3 to comment on the rebuttal to reviewer#2 and they felt the following two points required further action:

- "Fig. 3e. To generate the growth curves for the primary MLL-fusion and MLL wild type cells, the authors culture primary cells for 20-25 days. It is everyone's general experience that most primary AML cells die exponentially over a few days when placed in in vitro cell culture. Are special growth factors used to obtain survival and growth of the cells over 20-25 days? The authors need to show a real growth curve over this period for the control (shRen.713) cells. Do the control cells increase, decrease or stay the same in numbers over the 20-25 days? Could the authors compare Annexin V staining of control and SETD2 kd cells to see if either population is undergoing apoptosis?"- The heatmaps in figures 3d/6c need to be converted to real growth curves
- "Fig. 7 If SETD2 loss causes increased accumulation of DNA damage, then an inhibitor would cause genomic instability in the leukemic clone. Wouldn't this result in an increased rate of formation of resistant AML clones?" - This comment needs to be appropriately discussed in the text

Reviewer #4 (Remarks to the Author):

The authors have made an admirable effort to respond to the long list of concerns within the constraints of the model system they are using (ie, not using a conditional deletion version of

SetD2). Many sections of the study and manuscript have been clarified and improved.

There remains uncertainty regarding the differential effects of partial and complete loss of SetD2 in promoting/preventing tumor progression, and in part this is as the CRISPR approach failed to generate the correct full range of mutants (perhaps due selective disadvantage, as the authors). As raised by the latter referees, the mechanism for HOW SETD2 is interacting with MLL and mediating the various marks at H3K79 and K36 was, and is, unsatisfying in terms of mechanism.

It is helpful that direct interaction of MLL and SETD2 has been confirmed, although given the huge number of proteins that were pulled down, a more detailed confirmation of a short list of interacting proteins, and a more substantial effort to define the complex(es) in which SETD2 participates with MLL would have been helpful (and not beyond the scope of the manuscript).

We would like to thank all reviewers again for their additional and helpful comments. We carefully considered all of them and we are certain that the suggested changes help to better communicate our findings to the readers of *Nature Communications*. We are glad that our efforts in addressing the points raised during the previous round of revisions were recognized and appreciated. We hope that the current, further revised status of this manuscript will meet the expectations of the editorial board and the reviewers of *Nature Communications* and merit publication of our manuscript in this journal.

A point-by-point reply to the remarks of the reviewers is provided below.

The requested changes to the text are highlighted in yellow in the manuscript file.

Reviewer #3 (Remarks to the Author):

Review Skucha et al:

*This manuscript has been evolving since its last submission and it has improved. Yet, the whole gist of the manuscript still is not adequately adjusted to the novelty that is conveyed here.*

*The major points not properly acknowledged in this manuscript are as follows:*

*- It has been already shown that MLL fusions need the H3K36me binding (PWWP) domain of LEDGF for function (see PMID:18598942, and PMID:24465000 where the abstract reads: "Here we show that only two functional modules are necessary and sufficient for target recognition: those that bind to non-methylated CpGs and di-/tri-methylated histone H3 lysine 36 (H3K36me2/3).")*

*- A complete loss of Setd2 is NOT incompatible with leukemia development as the authors erroneously indicate. See Mar et al. (PMID:29018079) figure 3C. A clean, genetic deletion of Setd2 delays leukemogenesis but it does not block it completely.*

*Therefore I'd kindly advise the authors to modify their manuscript as follows (mostly text/display issues)*

*- Introduction, line 56: Please, correct your statement and acknowledge that it has been shown that menin is necessary to form a compound binding site (together with MLL) for LEDGF and that a direct fusion of the LEDGF-H3K36 reader domain (PWWP) to MLL can replace menin altogether.*

We have corrected this statement in the introduction of the manuscript. The importance of the PWWP-domain of LEDGF is now emphasized in the main text and the suggested references were included.

*- Introduction, line 78 to 82: " Thus, despite...". Delete this sentence as it is incorrect. Of course "shared mechanisms of transformations" have been demonstrated! What about the menin inhibitors?*

Following the reviewer's advice, we have deleted the sentence from the manuscript.

*- Results line 106: It needs to be acknowledged in the text and directly in the header of figure 1d that Jurkat T-cells were used here (write Jurkat, not "human leukemia line"). Do not detract from the fact that T-cells are a somewhat odd choice because T-ALL is almost never affected by 11q23 translocations. Also in line 108 it needs to be figure 1e not "1d".*

We did not intend to hide the cell line of our choice. We have adapted the main text as well as the corresponding figure to clearly state that the Jurkat leukemia cell line was used in this part of the study.

*- Results after line 130: A disclaimer is necessary here stating CLEARLY that most of the interactions identified will be indirect interactions. It is simply impossible from sterical considerations that MLL-N may interact directly with 128 different proteins.*

We appreciate this comment. Indeed, it is expected that many of identified interactors will not directly interact with MLL-fusions. A statement to clarify this has been introduced in the text.

*- Results line 173, suppl. figure 3c: Check your labeling. As displayed the logic of the control is wrong! The idea of the control is to show that a myc-specific precipitation will not unspecifically bring down V5-MLL in the absence of myc-Setd2. As shown the control doesn't make sense. Clearly, if V5-MLL is not there from the beginning it won't be precipitated. Either the experiment was set up the wrong way or the blots have been switched.*

We thank the reviewer for spotting this shortcoming. We have repeated the Western blots including all control samples. The results clearly show that Myc-antibodies do not cause any unspecific precipitation of the V5-tagged MLL construct. We have updated Supplementary Figure 3c in the revised version of the manuscript.

*- Results line 197: There's no "strong" anti-proliferative response. After three weeks the Setd2 k.d. cells still proliferate. A 1.5log difference in cell numbers after three weeks is significant but can be hardly called strong.*

We apologize for this overstatement. The wording was adapted.

*- Results line 198, fig 3d and similar: It is mandatory to show real data instead of "relative" heatmaps here. A simple line graph would be ok. It makes a difference if the GFP-positive ratio drops from 50% to 5% or from 50% to 40%. The heatmap only gives an undefined "max" and "min". I don't understand the refusal of the authors to show the actual data.*

To convince the reviewer from the reproducible anti-proliferative effect of SETD2 knockdown on MLL-fusion-expressing cells, we prepared a new subpanel in Supplementary Figure 4, which shows the actual percentages of GFP-positive cells over the course of the experiments. These data were used to generate the heatmap representation as shown in Figure 3d. We also now provide the same type of raw data that serve as basis for the heatmaps shown in Figure 6c in Supplementary Figure 9. We hope that together with the previously requested proliferation curves these additional datasets constitute a complementary and more convincing representation of our findings.

*- Results line 203, fig. 3e: Nothing can be really derived from a single NPMmut sample. It may be just a coincidence. In the absence of a solid and statistically valid group this is a mere chance observation. Please, either get more samples or take out this sub-figure.*

Following the reviewer's suggestion, the panel showing the results obtained from primary AML cells with NPM1c+ mutation has been removed. The manuscript text was updated accordingly.

*- Results line 220, fig.4c,d: Common consensus is that H3K79me2 and H3K36me3 are always and 100% correlated. The authors should show an additional example of a gene where this is not the case. Judging from the signal/noise ratio of the ChIP-seq results shown in fig. 4d, is it possible that for some genes the H3K36me/K79me signals disappear in the background and therefore the correlation is not complete?*

We would like to thank the reviewer for this remark. We have carefully reanalyzed our dataset and observed that the frequency of genes showing exclusive high H3K36me3 or H3K79me2 levels was indeed low. As pointed out, this might be caused by low signal-to-noise ratios at these genes. To communicate these findings more precisely, we now describe these genes as identified with “high” and “low” levels of the respective histone marks (instead of positive vs. negative, as shown previously) in Figure 4c. In addition, we provide additional examples for genes in our dataset that are marked by patterns of high H3K36me3 together with low H3K79me2 as well as low H3K36me3 together with high H3K79me2 in Supplementary Figure 5.

- Results line 265: *This would be a good place to insert a disclaimer that Setd2 knockout cells still can be transformed by MLL-AF9 although with a prolonged latency.*

A sentence indicating that knock-out of Setd2 greatly increased the latency of MLL-AF9-induced AML was introduced together with the respective reference (Mar et al., *Blood* 2017).

- Results line 289: *Change heading because Setd2 is not absolutely required for oncogenic transformation by MLL-fusion proteins. (see Mar et al).*

*Discussion: Please, check your discussion and take out any statement that would indicate that Setd2 is absolutely necessary for transformation by MLL fusions. Setd2 “accelerates” leukemogenesis but it is not essential as shown by genetically clean systems.*

We have rephrased the subtitle as well as all other parts of the manuscript, which might overstate the role of SETD2 in oncogenic transformation.

*In summary the authors do bring up a novel and interesting idea worth publishing. The fact that MLL should contact the very enzyme that deposits the modification that is read by MLL (through LEDGF) is new and very plausible. This situation is with precedent as the known MLL complex (COMPASS) member WDR5 also reads H3K4me that is deposited by wt-MLL. However, the fact that perturbation of H3K36me slows down transformation was highly anticipated as the importance of H3K36me for targeting MLL fusions has been prominently published before.*

*Editorial note: We asked Reviewer#3 to comment on the rebuttal to reviewer#2 and they felt the following two points required further action:*

•“Fig. 3e. To generate the growth curves for the primary MLL-fusion and MLL wild type cells, the authors culture primary cells for 20-25 days. It is everyone’s general experience that most primary AML cells die exponentially over a few days when placed in in vitro cell culture. Are special growth factors used to obtain survival and growth of the cells over 20-25 days? The authors need to show a real growth curve over this period for the control (shRen.713) cells. Do the control cells increase, decrease or stay the same in numbers over the 20-25 days? Could the authors compare Annexin V staining of control and SETD2 kd cells to see if either population is undergoing apoptosis?”- The heatmaps in figures 3d/6c need to be converted to real growth curves

As mentioned above, additional figures representing changes in the actual percentages of shRNA- and sgRNA-expressing cell populations over time have been included in the revised version of the manuscript (Supplementary Figure 4, Supplementary Figure 9)

•“Fig. 7 If SETD2 loss causes increased accumulation of DNA damage, then an inhibitor would cause genomic instability in the leukemic clone. Wouldn’t this result in an increased rate of formation of resistant AML clones?”- This comment needs to be appropriately discussed in the text

A statement indicating that the accumulation of DNA damage upon SETD2 loss might lead to accelerated outgrowth of chemo-resistant AML subclones has been included in the discussion section of the manuscript.

*Reviewer #4 (Remarks to the Author):*

*The authors have made an admirable effort to respond to the long list of concerns within the constraints of the model system they are using (ie, not using a conditional deletion version of SetD2). Many sections of the study and manuscript have been clarified and improved.*

*There remains uncertainty regarding the differential effects of partial and complete loss of SetD2 in promoting/preventing tumor progression, and in part this is as the CRISPR approach failed to generate the correct full range of mutants (perhaps due selective disadvantage, as the authors). As raised by the latter referees, the mechanism for HOW SETD2 is interacting with MLL and mediating the various marks at H3K79 and K36 was, and is, unsatisfying in terms of mechanism.*

We agree that a more detailed understanding of the dependency between H3K36me3 and H3K79me2 in the context of MLL-rearranged leukemia would provide valuable insight in the mechanism of SETD2 in this disease. However, as this manuscript reports the identification and validation of SETD2 as a critical effector of MLL-fusion proteins, we hope that the reviewer agrees that more mechanistic studies might represent interesting subjects for future work in this direction.

*It is helpful that direct interaction of MLL and SETD2 has been confirmed, although given the huge number of proteins that were pulled down, a more detailed confirmation of a short list of interacting proteins, and a more substantial effort to define the complex(es) in which SETD2 participates with MLL would have been helpful (and not beyond the scope of the manuscript).*

We agree that a deeper validation of protein-protein interactions of additional candidate proteins with MLL could shed additional light on the composition of protein complexes around MLL-fusion proteins. However, we were unable to perform additional work in this direction prior to resubmission of the manuscript for time reasons. But we would like to highlight that several shortlisted candidate genes in our network have been reported to be critical for MLL-rearranged AML by other groups. For instance, the protein SON was shown to interact with MEN1 and controls MLL-dependent transcription (Kim et al., *Mol Cell* 2016), which independently validates our approach. We emphasized this point in the discussion section of the manuscript.

REVIEWERS' COMMENTS:

Reviewer #3 (Remarks to the Author):

The authors have responded satisfactorily to most of my comments. As always, different viewpoints may persist in certain aspects. Yet, it would be unfair to delay publication further and to deny a wider audience the possibility to judge these findings on their own.